# FEDERATED LEARNING VIA PLURALITY VOTE

## ABSTRACT

Federated learning allows collaborative workers to solve a machine learning problem while preserving data privacy. Recent studies have tackled various challenges in federated learning, but the joint optimization of communication overhead, learning reliability, and deployment efficiency is still an open problem. To this end, we propose a new scheme named federated learning via plurality vote (FedVote). In each communication round of FedVote, workers transmit binary or ternary weights to the server with low communication overhead. The model parameters are aggregated via weighted voting to enhance the resilience against Byzantine attacks. When deployed for inference, the model with binary or ternary weights is resource-friendly to edge devices. We show that our proposed method can reduce quantization error and converges faster compared to the methods directly quantizing the model updates.

## 1    INTRODUCTION

Federated learning enables multiple workers to solve a machine learning problem under the coordination of a central server (Kairouz et al., 2021). Throughout the training stage, client data will be kept locally and only model weights or model updates will be shared with the server. Federated averaging (FedAvg) (McMahan et al., 2017) was proposed as a generic federated learning solution. Although FedAvg takes advantage of distributed client data while maintaining their privacy, it leaves the following two challenges unsolved. First, transmitting high-dimensional messages between a client and the server for multiple rounds can incur significant communication overhead. Quantization has been incorporated into federated learning in recent studies (Reisizadeh et al., 2020; Haddadpour et al., 2021). However, directly quantizing the gradient vector may not provide the optimal trade-off between communication efficiency and model accuracy [1]. Second, the aggregation rule in FedAvg is vulnerable to Byzantine attacks (Blanchard et al., 2017). Prior works tackled this issue by using robust statistics such as coordinate-wise median and geometric median in the aggregation step (Blanchard et al., 2017; Yin et al., 2018). Another strategy is to detect and reject updates from malicious attackers (Muñoz-González et al., 2019; Sattler et al., 2020). The robustness of the algorithm is enhanced at the cost of additional computation and increased complexity of algorithms.

In this paper, we propose a new method called federated learning via plurality vote (FedVote). We train a neural network at each worker with a range normalization function applied to model parameters. After local updating, binary/ternary weight vectors are obtained via stochastic rounding and sent to the server. The global model is updated by a voting procedure, and the voting results are sent back to each worker for further optimization in the next round. The contributions of the paper are summarized as follows.

1. We present FedVote as a novel federated learning solution to jointly optimize the communication overhead, learning reliability, and deployment efficiency.

2. We theoretically and experimentally verify the effectiveness of our FedVote design. In bandwidth-limited scenarios, FedVote is particularly advantageous in simultaneously achieving a high compression ratio and good test accuracy. Given a fixed communication cost, FedVote improves model accuracy on the CIFAR-10 dataset by 5–10%, 15–20%,

---

[1] Predictive coding in video and image compression (Li et al., 2015; Gonzalez & Woods, 2014) is an example that directly quantizing the raw signal we intend to transmit does not provide the best trade-off between the coding efficiency and the utility/bitrate.

and 25–30% compared with FedPAQ (Reisizadeh et al., 2020), signSGD (Bernstein et al., 2018), and FedAvg, respectively.

3. We extend FedVote to incorporate reputation-based voting. The proposed method, Byzantine-FedVote, exhibits much better resilience to Byzantine attacks in the presence of close to half attackers without incurring excessive computation compared with existing algorithms.

## 2 RELATED WORK

**Communication-Efficient Federated Learning.** In the prior study of federated learning, various strategies have been proposed to reduce communication cost. One research direction is to reduce the size of messages in each round. For example, Bernstein et al. (2018) showed that sign-based gradient descent schemes can converge well in the homogeneous data distribution scenario, while Chen et al. (2020); Jin et al. (2020); Safaryan & Richtarik (2021) extended it to the heterogeneous data distribution setting. In parallel, FedAvg adopts a periodic averaging scheme and targets at reducing the number of communication rounds (McMahan et al., 2017). Hybrid methods consider simultaneous local updates and accumulative gradient compression (Reisizadeh et al., 2020; Haddadpour et al., 2021). In this work, we improve communication efficiency by employing binary/ternary weights in the neural network.

**Quantized Neural Networks.** Quantized neural networks aim to approximate the full-precision networks using quantized weights while keeping their generalizability. As a special case, binary neural networks (BNNs) are gaining popularity in recent years. By restricting model weight values to $\{-1, +1\}$, BNNs can reduce computational cost, memory requirement, and energy consumption. Hubara et al. (2016) introduced real-valued latent weights and used the sign operator for binarization. In contrast, Shayer et al. (2018) let the neural network learn the distribution of the binary or ternary weights. Gong et al. (2019) added a soft quantization function to the real-valued weights, thus avoiding the gradient mismatch between the forward and the backward passes. A more thorough survey on BNN optimization can be found in Qin et al. (2020a).

**Distributed Optimization of Quantized Neural Networks.** A few recent works have explored BNN optimization in a distributed setting, which is more relevant to our work. Lin et al. (2020) conducted the case study of 1-bit quantized local models aggregated via ensemble distillation. The aggregation becomes complicated due to the separate optimization stage of knowledge distillation, especially when the distillation algorithm does not converge well in practice. In addition, their BNN optimization is not tailored to the federated learning setting. Hou et al. (2019) theoretically analyzed the convergence of distributed quantized neural networks, which was later implemented in the application of intrusion detection (Qin et al., 2020b). In comparison, FedVote is presented as a new algorithm by leveraging client local updates to accelerate the training. Different from Hou et al. (2019), we do not assume a convex and twice differentiable objective function and bounded gradients. Therefore, the analyses in Hou et al. (2019) cannot be directly applied to our study.

**Byzantine Resilience in Federated Learning.** An attack is Byzantine if arbitrary outputs are produced due to the adversary (Kairouz et al., 2021). Blanchard et al. (2017) showed that FedAvg cannot tolerate a single Byzantine attacker. They proposed an aggregation-rule-based remedy using the similarity of local updates. Similarly, Yin et al. (2018) took the advantage of coordinatewise median and trimmed-mean to robustify the aggregation. Muñoz-González et al. (2019) and Sattler et al. (2020) detected the adversaries and filtered out their model updates. In this paper, we propose a reputation-based voting strategy for FedVote that is shown to have good convergence performance in the Byzantine setting.

## 3 PRELIMINARIES

Symbol conventions are as follows. Bold lower cases of letters such as $\boldsymbol{v}_m$ denote column vectors, and $v_{m,i}$ is used to denote its $i$th entry. For a scalar function, it applies elementwise operation when a vector input is given. $\mathbf{1} = [1, \ldots, 1]^\top$ denotes a vector with all entries equal to 1.

### 3.1 Federated Learning

The goal of federated learning is to build a machine learning model based on the training data distributed among multiple workers. To facilitate the learning procedure, a server will coordinate the training without seeing the raw data (Kairouz et al., 2021). In a supervised learning scenario, let $\mathcal{D}_m = \{(\mathbf{x}_{m,j}, \mathbf{y}_{m,j})\}_{j=1}^{n_m}$ denote the training dataset on the $m$th worker, with the input $\mathbf{x}_{m,j} \in \mathbb{R}^{d_1}$ and the label $\mathbf{y}_{m,j} \in \mathbb{R}^{d_2}$ in each training pair. The local objective function $f_m$ with a model weight vector $\boldsymbol{\theta} \in \mathbb{R}^d$ is given by

$$f_m(\boldsymbol{\theta}) \triangleq f_m(\boldsymbol{\theta}; \mathcal{D}_m) = \frac{1}{n_m} \sum_{j=1}^{n_m} \ell(\boldsymbol{\theta}; (\mathbf{x}_{m,j}, \mathbf{y}_{m,j})), \tag{1}$$

where $\ell$ is a loss function quantifying the error of model $\boldsymbol{\theta}$ predicting the label $\mathbf{y}_{m,j}$ for an input $\mathbf{x}_{m,j}$. A global objective function may be formulated as

$$\min_{\boldsymbol{\theta} \in \mathbb{R}^d} f(\boldsymbol{\theta}) = \frac{1}{M} \sum_{m=1}^{M} f_m(\boldsymbol{\theta}). \tag{2}$$

### 3.2 Quantized Neural Networks

Consider a neural network $g$ with the weight vector $\boldsymbol{\theta} \in \mathbb{R}^d$. A forward pass for an input $\mathbf{x} \in \mathbb{R}^{d_1}$ and a prediction $\hat{\mathbf{y}} \in \mathbb{R}^{d_2}$ can be written as $\hat{\mathbf{y}} = g(\boldsymbol{\theta}, \mathbf{x})$. In quantized neural networks, the real-valued $\boldsymbol{\theta}$ is replaced by $\mathbf{w} \in \mathbb{D}_n^d$, where $\mathbb{D}_n$ is a discrete set with a number $n$ of quantization levels. For example, we may have $\mathbb{D}_2 = \{-1, 1\}$ for a binary neural network. For a given training set $\{(\mathbf{x}_j, \mathbf{y}_j)\}_{j=1}^{N}$ and the loss function $\ell$, the goal is to find an optimal $\mathbf{w}^*$ such that the averaged loss is minimized over a search space of quantized weight vectors:

$$\mathbf{w}^\star = \operatorname*{argmin}_{\mathbf{w} \in \mathbb{D}_n^d} \frac{1}{N} \sum_{j=1}^{N} \ell(\mathbf{w}; (\mathbf{x}_j, \mathbf{y}_j)). \tag{3}$$

Prior studies tried to solve (3) by optimizing a real-valued latent weight vector $\boldsymbol{h} \in \mathbb{R}^d$ (Hubara et al., 2016; Shayer et al., 2018; Gong et al., 2019). The interpretations of the latent weight vary when viewed from different perspectives. Hubara et al. (2016) used the sign operation to quantize the latent weight into two levels during the forward pass. The binary weights can be viewed as an approximation of their latent real-valued counterparts. Shayer et al. (2018) trained a stochastic binary neural network, and the normalized latent parameters are interpreted as the Bernoulli distribution parameter $\vartheta_i$:

$$\vartheta_i \triangleq \widehat{\mathbb{P}}(w_i = 1) = S(h_i), \quad w_i \in \{0, 1\}, \tag{4}$$

where $S : \mathbb{R} \to (0, 1)$ is the sigmoid function. In the forward pass, instead of using the binary vector $\mathbf{w}$, its expected value,

$$\widetilde{\mathbf{w}}_{\text{sto-bnn}} \triangleq \mathbb{E}[\mathbf{w}] = -1 \times [\mathbf{1} - S(\boldsymbol{h})] + 1 \times S(\boldsymbol{h}) = 2\,S(\boldsymbol{h}) - \mathbf{1}, \tag{5}$$

will participate in the actual convolution or matrix multiplication operations. In other words, the neural network function becomes $\hat{\mathbf{y}} = g(\widetilde{\mathbf{w}}_{\text{sto-bnn}}, \mathbf{x})$. Likewise, Gong et al. (2019) normalized the latent weight but interpreted it from a different viewpoint. They approximated the staircase quantization function with a differentiable soft quantization (DSQ) function, i.e.,

$$\widetilde{\mathbf{w}}_{\text{dsq}} \triangleq \tanh(a\boldsymbol{h}), \tag{6}$$

where $\tanh : \mathbb{R} \to (-1, 1)$ is the hyperbolic tangent function, and $a$ controls the shape of the function. The neural network function thus becomes $\hat{\mathbf{y}} = g(\widetilde{\mathbf{w}}_{\text{dsq}}, \mathbf{x})$.

We now summarize the latent-weight-based BNN training methods and depict an example of a single-layer network in Figure 1. First, a real-valued vector $\boldsymbol{h} \in \mathbb{R}^d$ is introduced and its range is restricted using a differentiable and invertible normalization function $\varphi : \mathbb{R} \to (-1, 1)$. The forward pass is then calculated with the normalized weight vector $\widetilde{\mathbf{w}}$. The procedure is described as:

$$\hat{\mathbf{y}} = g(\widetilde{\mathbf{w}}, \mathbf{x}), \quad \widetilde{\mathbf{w}} \triangleq \varphi(\boldsymbol{h}). \tag{7}$$

Second, in the back propagation, the latent weight vector $\boldsymbol{h}$ is updated with its gradient, i.e., $\boldsymbol{h}^{(t+1)} = \boldsymbol{h}^{(t)} - \eta \nabla_{\boldsymbol{h}} \ell$. Finally, the normalized weight vector $\widetilde{\mathbf{w}}$ are mapped to the discrete space to approximate $\mathbf{w}^*$ via thresholding or stochastic rounding.

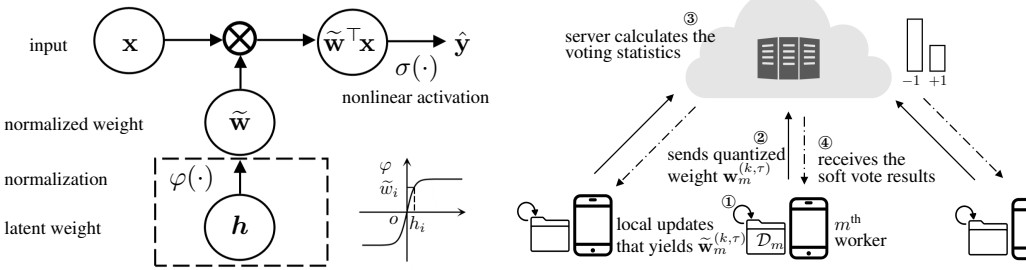

Figure 1: Example of a single-layer quantized neural network with a latent weight vector $\boldsymbol{h} \in \mathbb{R}^d$. $\boldsymbol{h}$ is normalized to generate $\widetilde{\mathbf{w}} \in (-1, 1)^d$, and the output is $\hat{\mathbf{y}} = \sigma(\widetilde{\mathbf{w}}^\top \mathbf{x})$. A binary weight $\mathbf{w}_q$ can be obtained by thresholding or rounding $\widetilde{\mathbf{w}}$ to the discrete space $\mathbb{D}_2^d$.

Figure 2: One round of FedVote is composed of four steps. Each worker first updates the local model and then sends the quantized weight $\mathbf{w}_m^{(k,\tau)}$ to the server. Later, the server calculates the voting statistics and sends back the soft voting results $\boldsymbol{p}^{(k+1)}$ to each worker.

## 4 PROPOSED FEDVOTE ALGORITHM

In this section, we present our proposed method with an emphasis on uplink communication efficiency and enhanced Byzantine resilience. We follow the widely adopted analysis framework in wireless communication to investigate only the worker uplink overhead, assuming that the downlink bandwidth is much larger and the server will have enough transmission power (Tran et al., 2019). To reduce the message size per round, we train a quantized neural network under the federated learning framework. The goal is to find a quantized weight vector $\mathbf{w}^*$ that minimizes the global objective function $f$ formulated in (2), i.e.,

$$\mathbf{w}^* = \underset{\mathbf{w} \in \mathbb{D}_n^d}{\arg\min} f(\mathbf{w}). \tag{8}$$

For the simplicity of presentation, we mainly focus on the BNN case with $\mathbb{D}_2 = \{-1, 1\}$. We illustrate the procedure in Figure 2 and provide the pseudo code in Appendix B. Below, we explain each step in more detail.

### 4.1 LOCAL MODEL TRAINING AND TRANSMISSION

We optimize a neural network with a learnable latent weight vector $\boldsymbol{h}$. In the $k$th communication round, we assume all workers are identically initialized by the server, namely, $\forall\, m \in \{1, \ldots, M\}$, $\boldsymbol{h}_m^{(k,0)} = \boldsymbol{h}^{(k)}$. To reduce the total number of communication rounds, we first let each worker conduct local updates to learn the binary weights. For each local iteration step, the local latent weight vector $\boldsymbol{h}^{(k,t+1)}$ is updated by the gradient descent:

$$\boldsymbol{h}_m^{(k,t+1)} = \boldsymbol{h}_m^{(k,t)} - \eta\, \nabla_{\boldsymbol{h}} f_m \left( \varphi(\boldsymbol{h}_m^{(k,t)}); \xi_m^{(k,t)} \right), \quad t \in \{0, \ldots, \tau - 1\}, \tag{9}$$

where $\xi_m^{(k,t)}$ is a mini-batch randomly drawn from $\mathcal{D}_m$ at the $t$th iteration of round $k$. After updating for $\tau$ steps, we obtain $\boldsymbol{h}_m^{(k,\tau)}$ and the corresponding normalized weight vector $\widetilde{\mathbf{w}}_m^{(k,\tau)} \in (-1, 1)^d$ defined as follows:

$$\widetilde{\mathbf{w}}_m^{(k,\tau)} \triangleq \varphi(\boldsymbol{h}_m^{(k,\tau)}). \tag{10}$$

To reduce the message size, we use the stochastic rounding to draw a randomly quantized version $\mathbf{w}_m^{(k,\tau)}$ using $\widetilde{\mathbf{w}}_m^{(k,\tau)}$, namely,

$$w_{m,i}^{(k,\tau)} = \begin{cases} +1, & \text{with probability } \pi_{m,i}^{(k,\tau)} = \frac{1}{2} \left[ \widetilde{w}_{m,i}^{(k,\tau)} + 1 \right], \\ -1, & \text{with probability } 1 - \pi_{m,i}^{(k,\tau)}. \end{cases} \tag{11}$$

It can be shown that the stochastic rounding is an unbiased procedure, i.e., $\mathbb{E}[\mathbf{w}_m^{(k,\tau)} | \widetilde{\mathbf{w}}_m^{(k,\tau)}] = \widetilde{\mathbf{w}}_m^{(k,\tau)}$. After quantization, the local worker will send the binary weights $\mathbf{w}_m^{(k,\tau)}$ to the server for the global model aggregation.

## 4.2 GLOBAL MODEL AGGREGATION AND BROADCAST

Once the server gathers the binary weights from all workers, it will perform the aggregation via plurality vote, i.e., $\mathbf{w}^{(k+1)} = \text{sign}\left(\sum_{m=1}^{M} \mathbf{w}_m^{(k,\tau)}\right)$. A tie in vote will be broken randomly. In the following lemma, we show that the probability of error reduces exponentially as the number of workers increases. The proof can be found in Appendix D.1.

**Lemma 1** *(One-Shot FedVote) Let $\mathbf{w}^* \in \mathbb{D}_2^d$ be the optimal solution defined in (8). For the $m$th worker, $\varepsilon_{m,i} \triangleq \mathbb{P}(w_{m,i}^{(k,\tau)} \neq w_i^*)$ denotes the error probability of the voting result of the $i$th coordinate. Suppose the error events $\{w_{m,i}^{(k,\tau)} \neq w_i^*\}_{m=1}^{M}$ are mutually independent, and the mean error probability $s_i = \frac{1}{M}\sum_{m=1}^{M} \varepsilon_{m,i}$ is smaller than $\frac{1}{2}$. For the voted weight $\mathbf{w}^{(k+1)}$, we have*

$$\mathbb{P}\left(w_i^{(k+1)} \neq w_i^*\right) \leqslant \left[2s_i \exp(1 - 2s_i)\right]^{\frac{M}{2}}. \tag{12}$$

In practice, the number of available workers may be limited in each round, and the local data distribution is often heterogenous or even time-variant. Therefore, it is almost always desirable to execute FedVote in a multiple-round fashion. In this case, we first use the soft voting to build an empirical distribution of global weight $\mathbf{w}$, i.e.,

$$\widehat{\mathbb{P}}(w_i^{(k+1)} = 1) = \frac{1}{M}\sum_{m=1}^{M} \mathbb{1}\left(w_{m,i}^{(k,\tau)} = 1\right), \tag{13}$$

where $\mathbb{1}(\cdot) \in \{0, 1\}$ is the indicator function. Let $p_i^{(k+1)} \triangleq \widehat{\mathbb{P}}(w_i^{(k+1)} = 1)$ and $\boldsymbol{p}^{(k+1)} = [p_1^{(k+1)}, \ldots, p_d^{(k+1)}]^\top$. The global latent parameters can be constructed by following (10):

$$\boldsymbol{h}^{(k+1)} = \varphi^{-1}(2\boldsymbol{p}^{(k+1)} - 1), \tag{14}$$

where $\varphi^{-1} : (-1, 1) \to \mathbb{R}$ is the inverse of the normalization function $\varphi$. We further apply clipping to restrict the range of the probability, namely, $\text{clip}(p_i^{(k+1)}) = \max(p_{\min}, \min(p_{\max}, p_i^{(k+1)}))$, where $p_{\min}, p_{\max} \in (0, 1)$ are predefined thresholds. To keep the notation consistent, we denote $\widetilde{\mathbf{w}}^{(k+1)} \triangleq \varphi(\boldsymbol{h}^{(k+1)})$ as the global normalized weight. After broadcasting the soft voting results $\boldsymbol{p}^{(k+1)}$, all workers are synchronized with the same latent weight $\boldsymbol{h}^{(k+1)}$ and normalized weight $\widetilde{\mathbf{w}}^{(k+1)}$. The learning procedure will repeat until a termination condition is satisfied. We relate FedVote to FedAvg in the following lemma. The detailed proof can be found in Appendix D.2.

**Lemma 2** *(Relationship with FedAvg) For the normalized weights, FedVote recovers FedAvg in expectation:* $\mathbb{E}\left[\widetilde{\mathbf{w}}^{(k+1)}\right] = \frac{1}{M}\sum_{m=1}^{M} \widetilde{\mathbf{w}}_m^{(k,\tau)}$, *where* $\widetilde{\mathbf{w}}^{(k+1)} = \varphi(\boldsymbol{h}^{(k+1)})$ *and* $\widetilde{\mathbf{w}}_m^{(k,\tau)} = \varphi(\boldsymbol{h}_m^{(k,\tau)})$.

## 4.3 REPUTATION-BASED BYZANTINE-FEDVOTE

Lemma 2 shows that FedVote is related to FedAvg in expectation. As we have reviewed in Section 2, FedAvg cannot tolerate a single Byzantine attacker. It indicates that FedVote will exhibit similar poor performance in the presence of multiple adversaries (see Appendix C.2). We improve the design of FedVote based on a reputation voting mechanism, which in essence is a variant of the weighted soft voting method.

Reputation-based voting was presented in failure-robust large scale grids (Bendahmane et al., 2014). In our design, we modify (13) to $\widehat{\mathbb{P}}(w_i^{(k+1)} = 1) = \sum_{m=1}^{M} \lambda_m^{(k)} \mathbb{1}\left(w_{m,i}^{(k,\tau)} = 1\right)$, where $\lambda_m^{(k)}$ is proportional to a credibility score. In Byzantine-resilient FedVote (Byzantine-FedVote), we assume that at least 50% of the workers behave normally and treat the plurality vote result as the correct decision. The credibility score of the $m$th worker is calculated by counting the number of correct votes it makes: $\text{CR}_m^{(k+1)} = \frac{1}{d}\sum_{i=1}^{d} \mathbb{1}\left(w_{m,i}^{(k,\tau)} = w_i^{(k+1)}\right)$. Through multiple rounds, we track the credibility of a local worker by taking an exponential moving average over the communication rounds, namely, $\nu_m^{(k+1)} = \beta \nu_m^{(k)} + (1 - \beta)\text{CR}_m^{(k+1)}$, where $\beta \in (0, 1)$ is a predefined coefficient. The weight $\lambda_m^{(k)}$ is designed as $\lambda_m^{(k)} = \nu_m^{(k)}/\sum_{m=1}^{M} \nu_m^{(k)}$.

Even though the presentation in this section focuses on the binary weights, the scheme can be naturally extended to quantized neural networks of more discrete levels. We will briefly discuss the implementation for ternary weights in Section 6.

## 5 ANALYSIS OF ALGORITHM

In this section, we present the theoretical analysis of FedVote when local data are independent and identically distributed (i.i.d.). The empirical results in the non-i.i.d. setting will be discussed in Section 6, and the corresponding analyses are left for future work. We use the gradient norm expectation as an indicator of convergence, which is commonly adopted in nonconvex optimization literature (Reisizadeh et al., 2020; Haddadpour et al., 2021). To simplify the notation, we first denote the stochastic local gradient by $\tilde{\boldsymbol{g}}_m^{(k,t)} \triangleq \nabla_{\boldsymbol{h}} f_m(\varphi(\boldsymbol{h}_m^{(k,t)}); \xi_m^{(k,t)})$. The local true gradient and global true gradient will be denoted by $\boldsymbol{g}_m^{(k,t)} \triangleq \mathbb{E}_\xi[\tilde{\boldsymbol{g}}_m^{(k,t)}]$, $\boldsymbol{g}^{(k)} \triangleq \nabla_{\boldsymbol{h}} f(\varphi(\boldsymbol{h}^{(k)}))$, respectively. In addition, let $\boldsymbol{\zeta}_m^{(k)} \triangleq \widetilde{\mathbf{w}}_m^{(k,\tau)} - \mathbf{w}_m^{(k,\tau)}$ denote the error introduced by stochastic rounding. According to the unbiased property of stochastic rounding, we have $\mathbb{E}_{\boldsymbol{\pi}}[\boldsymbol{\zeta}_m^{(k)}] = \boldsymbol{0}$. With the aforementioned notations, we state five assumptions for the convergence analysis.

### 5.1 ASSUMPTIONS

**Assumption 1** *(Lower bound)* $\forall \ \boldsymbol{h} \in \mathbb{R}^d$, $\widetilde{\mathbf{w}} \in (-1, 1)^d$, *the objective function is lower bounded by a constant* $f^*$, *i.e.,* $f(\widetilde{\mathbf{w}}) \geqslant f^* = \min_{\boldsymbol{h} \in \mathbb{R}^d} f(\varphi(\boldsymbol{h}))$.

**Assumption 2** *(L-smoothness)* $\forall \ \widetilde{\mathbf{w}}_1, \widetilde{\mathbf{w}}_2 \in (-1, 1)^d$, $m \in \{1, \dots, M\}$, *there exists some non-negative* $L$ *such that* $\|\nabla f_m(\widetilde{\mathbf{w}}_1) - \nabla f_m(\widetilde{\mathbf{w}}_2)\|_2 \leqslant L \|\widetilde{\mathbf{w}}_1 - \widetilde{\mathbf{w}}_2\|_2$.

Assumptions 1 to 2 are common for necessary analyses (Wang & Joshi, 2018). We limit the range of the normalized weight $\widetilde{\mathbf{w}}$ while in a typical setting there is no restriction to the model weight.

**Assumption 3** *The normalization function* $\varphi : \mathbb{R} \to (-1, 1)$ *is strictly increasing. In particular, we assume its first derivative is bounded for all* $h_{m,i}^{(k,t)}$, *i.e.,* $\frac{\mathrm{d}}{\mathrm{d}h}\varphi(h_{m,i}^{(k,t)}) \in [c_1, c_2]$, *where* $c_1$, $c_2$ *are positive parameters independent of* $k$, $t$, $m$, *and* $i$.

Assumption 3 is not difficult to satisfy in practice. For example, let $\varphi(h) = \tanh(ah)$, we have $\varphi'(h) = a\left[1 - \tanh^2(ah)\right]$, with $c_2 = a$. Note that $\varphi$ quickly saturates with a large $h$ in the local updating. On the other hand, the empirical Bernoulli parameter $p_i$ will be clipped for stability, which indicates that $h_{m,i}^{(k,t)}$ will be upper bounded by certain $h_{\mathrm{B}}$. In this sense, we have $c_1 = a\left[1 - \tanh^2(ah_{\mathrm{B}})\right]$. The next two assumptions bound the variance of the stochastic gradient and quantization noise.

**Assumption 4** *The stochastic gradient has bounded variance, i.e.,* $\mathbb{E}\big\|\boldsymbol{g}_m^{(k,t)} - \tilde{\boldsymbol{g}}_m^{(k,t)}\big\|_2^2 \leqslant \sigma_\varepsilon^2$, *where* $\sigma_\varepsilon^2$ *is a fixed variance independent of* $k$, $t$ *and* $m$.

**Assumption 5** *The quantization error* $\boldsymbol{\zeta}_m^{(k)}$ *has bounded variance, i.e.,* $\mathbb{E}\big\|\boldsymbol{\zeta}_m^{(k)}\big\|_2^2 \leqslant \sigma_k^2$, *where* $\sigma_k^2$ *is a fixed variance independent of* $m$.

Note that quantization error is affected by the quantizer type and the corresponding input. For example, if the normalization function $\varphi$ approximates the sign function very well, the stochastic quantization error will be close to zero. Formally, the upper bound $\sigma_\zeta^2$ can be viewed as a function of input dimension $d$, which we formulate in the following lemma. The proof is in Appendix D.3.

**Lemma 3** *Suppose we have an input* $\boldsymbol{a} \in (-1, 1)^d$ *for the quantizer* $Q_{\mathrm{sr}}$ *defined in (11), then the quantization error satisfies* $\mathbb{E}\left[\big\|Q_{\mathrm{sr}}(\boldsymbol{a}) - \boldsymbol{a}\big\|_2^2 \big| \boldsymbol{a}\right] = d - \|\boldsymbol{a}\|_2^2$.

For existing algorithms quantizing the model update $\boldsymbol{\delta}_m^{(k)} \triangleq \boldsymbol{\theta}^{(k)} - \boldsymbol{\theta}_m^{(k,\tau)}$, the quantizer has the property $\mathbb{E}\left[\big\|Q(\mathbf{x}) - \mathbf{x}\big\|_2^2 \big| \mathbf{x}\right] \leqslant q\|\mathbf{x}\|_2^2$. With a fixed quantization step, $q$ increases when the input dimension $d$ increases (Basu et al., 2020). We state the result for a widely-used quantizer, QSGD (Alistarh et al., 2017), which has been adopted in FedPAQ, in the following lemma.

**Lemma 4** *Suppose we have an input* $\mathbf{x} \in \mathbb{R}^d$ *for the QSGD quantizer* $Q$. *In the coarse quantization scenario, the quantization error satisfies* $\mathbb{E}\left[\big\|Q(\mathbf{x}) - \mathbf{x}\big\|_2^2 \big| \mathbf{x}\right] = O\left(d^{\frac{1}{2}}\right) \|\mathbf{x}\|_2^2$.

## 5.2 CONVERGENCE ANALYSIS

We state the convergence results in the following theorem. The proof can be found in Appendix D.5.

**Theorem 1** *For FedVote under Assumptions 1 to 5, let the learning rate $\eta = O\left(\left(\frac{c_1}{c_2}\right)^2 \frac{1}{L\tau\sqrt{K}}\right)$, then after $K$ rounds of communication, we have*

$$\frac{1}{K}\sum_{k=0}^{K-1} c_1^2 \mathbb{E}\|\nabla f(\widetilde{\mathbf{w}}^{(k)})\|_2^2 \leqslant \frac{2\left[f(\widetilde{\mathbf{w}}^{(0)}) - f(\widetilde{\mathbf{w}}^*)\right]}{\eta\tau K} + c_2^2 L\eta \left[\frac{1}{M} + \frac{c_1^2 L\eta(\tau-1)}{2}\right]\sigma_\varepsilon^2$$

$$+ \frac{L}{\eta\tau KM}\sum_{k=0}^{K-1}\sigma_k^2 + \frac{2(c_2^2 - c_1^2)}{\tau MK}\sum_{k=0}^{K-1}\sum_{m=1}^{M} R_m^{(k)}. \tag{15}$$

*where $R_m^{(k)} \triangleq -\sum\limits_{t=0}^{\tau-1}\sum\limits_{i\notin\mathcal{I}_m^{(k,t)}} \mathbb{E}\left[(\nabla f(\widetilde{\mathbf{w}}^{(k)}))_i (\nabla f(\widetilde{\mathbf{w}}_m^{(k,t)}))_i\right]$ and $\mathcal{I}_m^{(k,t)} \triangleq \left\{i \mid g_i^{(k)} g_{m,i}^{(k,t)} \geqslant 0\right\}$.*

**Remark 1** *When there is no normalization function and quantization, i.e., $\varphi(x) = x$ with $c_1 = c_2 = 1$, $\sigma_k^2 = 0$, Theorem 1 recovers the result obtained in Wang & Joshi (2018).*

To discuss the impact of quantization error, consider the distribution of different inputs. For the model update $\boldsymbol{\delta}_m^{(k)}$, we expect the central limit theorem to render its distribution shape, where each entry $\delta_{m,i}^{(k)}$ follows the Gaussian distribution. For the Bernoulli probability $\boldsymbol{\pi}_m^{(k,\tau)}$, we expect the Beta distribution as the conjugate prior to render its distribution shape, where each entry $\pi_{m,i}^{(k,\tau)}$ follows the symmetric Beta distribution. See Figure 3 for the empirical results.

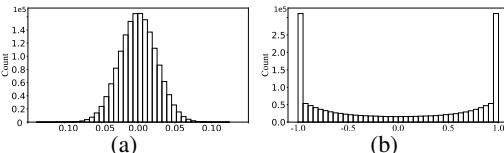
(a)    (b)

Figure 3: Histograms of (a) model updates $\delta_{m,i}^{(k)}$ and (b) binary weight probabilities $\pi_{m,i}^{(k,\tau)}$. We trained a LeNet on MNIST for a single communication round.

**Remark 2** *Following the analysis framework in Theorem 1, the order-wise convergence rate is $\frac{1}{K}\sum_{k=0}^{K-1}\mathbb{E}\left\|\nabla f\left(\mathbf{w}^{(k)}\right)\right\|_2^2 = \mathcal{O}(\frac{1}{\sqrt{K}}) + E(d)$, where $E(d)$ is the error introduced by the quantization. For FedVote with the normalized weight $\widetilde{\mathbf{w}}_m^{(k,\tau)}$ as the input, when $\pi_{m,i}^{(k,\tau)}$'s follow the symmetric Beta distribution, it can be shown that $\mathbb{E}\|\boldsymbol{\varsigma}_m^{(k)}\|_2^2 = O(d)$ based on Lemma 3. For algorithms such as FedPAQ with the model update $\boldsymbol{\delta}_m^{(k)}$ as the input, when $\delta_{m,i}^{(k)}$'s follow the Gaussian distribution, it can be shown that $\mathbb{E}\|Q(\boldsymbol{\delta}_m^{(k)}) - \boldsymbol{\delta}_m^{(k)}\|_2^2 = O(d^{3/2})$ based on Lemma 4. When the weight dimension $d$ is sufficiently large, FedVote converges faster.*

**Remark 3** *The value of the positive scalar error term $R_m^{(k)}$ in (15) depends on the gradient dissimilarity. If the angle between the local gradient $\nabla f_m(\widetilde{\mathbf{w}}_m^{(k,t)})$ and the global gradient $\nabla f(\widetilde{\mathbf{w}}^{(k,t)})$ is not large, $R_m^{(k)}$ can be treated as a bounded variable.*

**Remark 4** *The choice of nonlinear function $\varphi : \mathbb{R}^d \to (-1, 1)^d$ will affect the convergence. If $\varphi$ behaves more like the $\mathrm{sign}(\cdot)$ function, e.g., when $a$ increases in $\tanh(ax)$, the quantization error will be reduced. In other words, we expect a smaller $\sigma_k^2$ according to Lemma 3, which leads to a tighter bound in (15). Meanwhile, a larger $c_2$ will negatively influence the convergence.*

## 6 EXPERIMENTAL RESULTS

**Data and Models.** We choose image classification datasets Fashion-MNIST (Xiao et al., 2017) and CIFAR-10 (Krizhevsky, 2009). Both of them have a total of $C = 10$ classes. We consider two data partition strategies: (i) i.i.d. setting where the whole dataset is randomly shuffled and assigned to each worker without overlap; (ii) non-i.i.d. setting where we follow Hsu et al. (2019) and use the Dirichlet distribution to simulate the heterogeneity. In particular, for the $m$th worker we draw a random vector $\boldsymbol{q}_m \sim \mathrm{Dir}(\alpha)$, where $\boldsymbol{q}_m = [q_{m,1}, \cdots, q_{m,C}]^\top$ belongs to the standard $(C-1)$-simplex. We then assign data samples from different classes to the $m$th worker following the distribution of

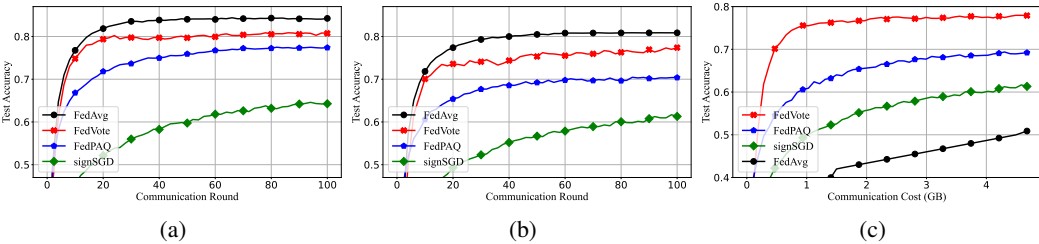

Figure 4: Learning curves of different methods on CIFAR-10 with (a) the i.i.d. setting and (b) the Dirichlet non-i.i.d. setting. Compared with the gradient quantization methods such as signSGD (Bernstein et al., 2018) and the model update quantization methods such as FedPAQ (Reisizadeh et al., 2020), FedVote achieves higher accuracy given the same number of communication rounds. (c) Test accuracy versus accumulative uplink communication cost on the i.i.d. CIFAR dataset. Fed-Vote achieves the best accuracy given the transmitted data size.

Table 1: Test Accuracy in the Byzantine Setting

| Dataset | Distribution | signSGD | Co-Med | Proposed |
|---|---|---|---|---|
| Fashion-MNIST | i.i.d. | 61.5% | 77.7% | 91.4% |
| | non-i.i.d | 61.0% | 73.4% | 89.4% |
| CIFAR-10 | i.i.d. | 13.5% | 29.3% | 76.6% |
| | non-i.i.d | 11.0% | 28.7% | 72.0% |

Table 2: Effect of the Normalization Function

| Fashion-MNIST | | $a$ | | | |
|---|---|---|---|---|---|
| | | 0.5 | 1.5 | 2.5 | 10 |
| i.i.d. | float | 90.7% | 90.6% | 90.0% | 88.2% |
| | binary | 88.7% | 90.4% | 89.9% | 88.2% |
| non-i.i.d. | float | 87.3% | 86.9% | 85.7% | 85.0% |
| | binary | 83.3% | 85.5% | 85.2% | 84.6% |

$q_m$. We set $\alpha = 0.5$ unless noted otherwise. We use a LeNet-5 architecture for Fashion-MNIST and a VGG-7 architecture for CIFAR-10. Results are obtained over three repetitions.

**Implementation Details.** We provide implementation details in the proposed FedVote design. First, following prior works (Shayer et al., 2018; Gong et al., 2019), we keep the weights of the BNN final layer as floating-point values for the sake of the model performance. The weights of the final layer are randomly initialized with a shared seed and will be fixed during the training process. Second, we notice that for quantized neural networks, the batch normalization (BN) (Ioffe & Szegedy, 2015) after the convolutional layer is necessary to scale the activation. We use the static BN without learnable parameters and local statistics (Diao et al., 2021) to ensure the voting aggregation of binary weights. For the normalization function, we choose $\varphi(x) = \tanh(3x/2)$ unless noted otherwise. More details of the experimental setup can be found in Appendix C.1.

**Communication Efficiency and Convergence Rate.** In this experiment, we consider $M = 31$ workers with full participation and compare FedVote to different methods within $N = 100$ communication rounds. The results of partial participation can be found in Appendix C.3. The communication cost is calculated as the accumulative uplink message size from all workers. Figure 4 reveals that FedVote outperforms the gradient-quantization-based methods such as signSGD (Bernstein et al., 2018) that quantizes gradients to 1 bit signs, and FedPAQ (Reisizadeh et al., 2020) that quantizes the updates to 2 bits integers. Compared with FedPAQ, signSGD, and FedAvg, FedVote improves the test accuracy by 5–10%, 15–20%, and 25–30%, respectively, given the fixed communication costs of 1.5–4.7 GB.

**Byzantine Resilience.** This experiment validates the effectiveness of Byzantine-FedVote. We consider omniscient attackers who can access the datasets of normal workers and send the opposite of the aggregated results to the server. The number of attackers is 15, and the remaining 16 clients are normal workers. We compare the proposed method with coordinate-wise median based (CoMed) gradient descent (Yin et al., 2018) and signSGD (Bernstein et al., 2018), and report the testing accuracy after 100 communication rounds. We do not compare with FedAvg and FedPAQ, as they are fragile to Byzantine failure. The results are shown in Table 1. It can be observed that Byzantine-FedVote exhibits much better resilience to Byzantine attacks with close to half adversaries.

Table 3: Test Accuracy of TNN and BNN

| Dataset | Distri-bution | BNN | TNN |
|---|---|---|---|
| Fashion-MNIST | i.i.d. | 91.1% | 91.9% |
| | non-i.i.d | 88.3% | 89.4% |
| CIFAR-10 | i.i.d. | 80.5% | 82.5% |
| | non-i.i.d | 74.6% | 77.6% |

Table 4: Forward Pass Efficiency

| Neural Net | Weight Type | Adds | Muls | Energy (mJ) |
|---|---|---|---|---|
| LeNet-5 | float | $1.7 \times 10^9$ | $1.8 \times 10^9$ | 8.1 |
| | binary | $1.7 \times 10^9$ | $1.0 \times 10^5$ | 1.5 |
| VGG-7 | float | $4.8 \times 10^{10}$ | $5.4 \times 10^{10}$ | 242.9 |
| | binary | $4.8 \times 10^{10}$ | $2.1 \times 10^5$ | 43.3 |

**Normalization Function.** From Remark 4, we know that the normalization function can influence the model convergence. We empirically examine the impact in this experiment. For normalization function $\varphi(x) = \tanh(ax)$, we choose $a$ from $\{0.5, 1.5, 2.5, 10\}$. We test the model accuracy after 20 communication rounds on Fashion-MNIST. The results are shown in Table 2. As $a$ increases, the linear region of the normalization function shrinks, and the algorithm converges slower due to a larger $c_2$. On the other hand, the gap between the model with normalized weight $\widetilde{\mathbf{w}}$ and the one with binary weight $\mathbf{w}$ also decreases due to smaller quantization errors. A good choice of the normalization function should take this trade-off into consideration.

**Ternary Neural Network Extension.** In the previous sections, we focus on the BNNs. We extend FedVote to ternary neural networks (TNNs) and empirically verify its performance. Training and transmitting the categorical distribution parameters of the ternary weight may bring additional communication and computation cost to edge devices, we therefore simplify the procedure as follows. For each ternary weight $w_{m,i}^{(k,t)}$, we still keep a latent parameter $h_{m,i}^{(k,t)}$ to optimize. After $\tau$ local steps, we use the stochastic rounding to the normalized weight $\widetilde{\mathbf{w}}_m^{(k,\tau)} = \varphi(\boldsymbol{h}_m^{(k,\tau)})$ and obtain quantized weight $\mathbf{w}_m^{(k,\tau)}$. At the aggregation stage on the server, instead of calculating the vote distribution, we directly compute the global normalized weight as $\widetilde{\mathbf{w}}^{(k+1)} = \frac{1}{M} \sum_{m=1}^{M} \mathbf{w}_m^{(k,\tau)}$. More details can be found in Appendix C.4. The training results are shown in Table 3. As TNNs can further reduce the quantization error, their performance is better than the BNNs at the cost of additional 1 bit/dimension communication overhead.

**Deployment Efficiency.** We highlight the advantages of BNNs during deployment on edge devices. In FedVote, we intend to deploy lightweight quantized neural networks on the workers after the training procedure. BNNs require $32\times$ smaller memory size, which can save storage and energy consumption for memory access (Hubara et al., 2016). As we do not quantize the activations, the advantage of BNNs inference mainly lies in the replacement of multiplications by summations. Consider the matrix multiplication in a neural network with an input $\mathbf{x} \in \mathbb{R}^{d_1}$ and output $\mathbf{y} \in \mathbb{R}^{d_2}$: $\mathbf{y} = W^\top \mathbf{x}$. For a floating-point weight matrix $W \in \mathbb{R}^{d_1 \times d_2}$, the number of multiplications is $d_1 d_2$, whereas for a binary matrix $W_b \in \mathbb{D}_2^{d_1 \times d_2}$ all multiplication operations can be replaced by additions. We investigate the number of real multiplications and additions in the forward pass of different models and present the results in Table 4. We use the CIFAR-10 dataset and set the batch size to 100. As to the energy consumption calculation, we use 3.7 pJ and 0.9 pJ as in Hubara et al. (2016) for each floating-point multiplication and addition, respectively.

## 7 CONCLUSION

In this work, we have proposed FedVote to jointly optimize communication overhead, learning reliability, and deployment efficiency. In FedVote, the server aggregates neural networks with binary/ternary weights via voting. We have verified that FedVote can achieve good model accuracy even in coarse quantization settings. Compared with gradient quantization, model quantization is a more effective design that achieves better trade-offs between communication efficiency and model accuracy. With the voting-based aggregation mechanism, FedVote enjoys the flexibility to incorporate various voting protocols to increase the resilience against Byzantine attacks. We have demonstrated that Byzantine-FedVote exhibits much better Byzantine resilience in the presence of close to half attackers compared to the existing algorithms.

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

# A  VOTING METHODS

We review the definitions of several voting methods (Zhou, 2019). Suppose we have a set of $M$ individual voters and $C$ candidate output labels $\{\psi_1, \ldots, \psi_C\}$. For the $m$th voter, it will output result $\boldsymbol{v}_m = [v_{m,1}, \ldots, v_{m,C}]^\top$. Here, the $i$th entry $v_{m,i} \in \{0, 1\}$, which takes the value one if the voter chooses the candidate $\psi_i$ and zero otherwise.

*Majority vote* requires the winner to receive more than half of the votes; if none of the candidates receives more than half of the votes, a rejection option will be given. The majority vote result can be written as

$$\Psi_{\text{majority}} = \begin{cases} \psi_i, & \text{if } \sum_{m=1}^{M} v_{m,i} > \frac{1}{2} \sum_{j=1}^{C} \sum_{m=1}^{M} v_{m,j}, \\ \text{rejection, otherwise.} \end{cases} \tag{16}$$

*Plurality vote* takes the class label that receives the largest number of votes as the final winner, i.e.,

$$\Psi_{\text{plurality}} = \psi_{\underset{i}{\operatorname{argmax}} \sum_{m=1}^{M} v_{m,i}}. \tag{17}$$

In the binary case with $C = 2$, plurality vote resembles majority vote except that it does not have a rejection option. If more than two candidates receive the same number of votes, we randomly select one of them as the final results. *Weighted vote* assigns different weights for voters and the result can be written as

$$\Psi_{\text{weighted}} = \psi_{\underset{i}{\operatorname{argmax}} \sum_{m=1}^{M} \lambda_m v_{m,i}}, \tag{18}$$

where $\lambda_m$ is the weight assigned to the $m$th voter. In contrast to the aforementioned vote methods, *soft vote* produces the probability output. The probability that the $i$th candidate wins is calculated as

$$\widehat{\mathbb{P}}(\Psi_{\text{soft}} = \psi_i) = \frac{1}{M} \sum_{m=1}^{M} v_{m,i}. \tag{19}$$

# B  FEDVOTE ALGORITHM

We summarize the proposed FedVote method and its Byzantine-resilient variant in Algorithm 1. When there are no attackers involved, we use `Option I`. When we require the resilience against Byzantine failure, we choose `Option II`.

# C  SETUP AND ADDITIONAL EXPERIMENTS

## C.1  HYPERPARAMETERS

For the clipping thresholds, we set $p_{\min} = 0.001$ and $p_{\max} = 1 - p_{\min}$. The thresholds are introduced for numerical stability and have little impact on performance. We use $\beta = 0.5$ in Byzantine-FedVote. The choice of the smoothing factor $\beta$ has little impact on the final test accuracy, as the credibility score decays exponentially for adversaries over multiple communication rounds. We use the Adam optimizer and search using the learning rate $\eta$ over the set $\{10^{-4}, 3 \times 10^{-4}, 10^{-3}, 3 \times 10^{-3}, 10^{-2}, 3 \times 10^{-2}, 10^{-1}, 3 \times 10^{-1}\}$. We set the number of local iterations $\tau$ to 40 and the local batch size to 100.

## C.2  COMPARISON OF VANILLA FEDVOTE AND BYZANTINE-FEDVOTE

Lemma 2 shows that FedVote is related to FedAvg in expectation. Adversaries sending the opposite results will negatively affect the estimation of the weight distribution and impede the convergence in multiple rounds. We compare the test accuracy of Byzantine-FedVote, Vanilla FedVote, and signSGD on the non-i.i.d. CIFAR-10 dataset ($\alpha = 0.5$) with various numbers of omniscient attackers. Figure 6 reveals that the test accuracy of Vanilla FedVote drops severely when the number of adversaries increases, which is consistent with our analysis. In contrast, the drop of accuracy in Byzantine-FedVote is negligible, confirming its resilience to omniscient attackers.

---

**Algorithm 1:** Binary-Weight FedVote with/without Byzantine Tolerance

---

1    **initialize** $p^{(0)}$ and **broadcast**

2    **for** $k = 0, 1, \dots, N-1$ **do**

3       **on** $m^{\text{th}}$ **worker:**

4         **receive** $p^{(k)}$ **from** the server

5         **initialize** latent weight $h_m^{(k,0)} = \varphi^{-1}(2p^{(k)} - 1)$

6         **for** $t = 0 : \tau - 1$ **do**

7           $\tilde{g}_m^{(k,t)} = \nabla_h f_m(\varphi(h_m^{(k,t)}); \xi_m^{(t,r)})$

8           $h_m^{(k,t+1)} = h_m^{(k,t)} - \eta^{(k,t)}\, \tilde{g}_m^{(k,t)}$

9         $\widetilde{\mathbf{w}}_m^{(k,\tau)} = \varphi(h^{(k,\tau)})$

10         $\mathbf{w}_m^{(k,\tau)} = \text{sto\_rounding}(\widetilde{\mathbf{w}}_m^{(k,\tau)})$    ▷ Eq. (11)

11         **send** $\mathbf{w}_m^{(k,\tau)}$ **to** server

12       **on server:**

13         $\{\mathbf{w}^{(k+1)}, p^{(k+1)}\} = \text{vote}(\{\mathbf{w}_m^{(k,\tau)}\}_{m=1}^M)$

14         **broadcast** $p^{(k+1)}$ **to** workers

15    **function vote**$(\{\mathbf{w}_m^{(k,\tau)}\}_{m=1}^M)$

16       **for** $i = 1 : d$ **do**

17         $w_i^{(k+1)} = \text{sign}\left( \sum_{m=1}^M w_{m,i}^{(k,\tau)} \right)$

18         $p_i^{(k+1)} = \frac{1}{M} \sum_{m=1}^M \mathbb{1}\left( \widehat{w}_{m,i}^{(k,\tau)} = 1 \right)$    ▷ Option I

19         $p_i^{(k+1)} = \sum_{m=1}^M \lambda_m^{(k)} \mathbb{1}\left( w_{m,i}^{(k,\tau)} = 1 \right)$    ▷ Option II

20       **return** $\{\mathbf{w}^{(k+1)}, p^{(k+1)}\}$ to the server

---

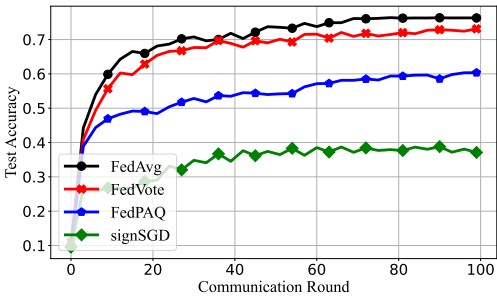 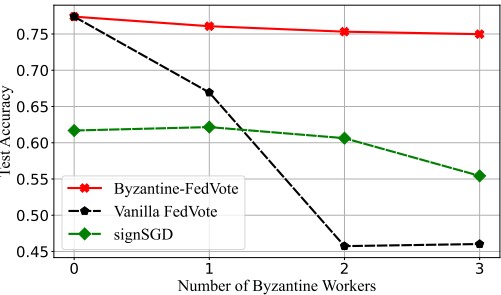

Figure 5: Test accuracy versus communication round of different methods on the non-i.i.d. CIFAR dataset ($\alpha = 0.5$). We use 100 workers and sample 20 of them in each round.

Figure 6: Test accuracy versus the number of Byzantine workers. As the number of adversaries increases, the test accuracy of FedVote drops rapidly.

### C.3   CONVERGENCE OF FEDVOTE WITH PARTIAL PARTICIPATION

We increase the number of workers to 100 and sample 20 of them in each communication round. We compare FedVote with FedAvg McMahan et al. (2017), FedPAQ (Reisizadeh et al., 2020), and signSGD (Bernstein et al., 2018) in Figure 5 on the non-i.i.d. CIFAR-10 dataset ($\alpha = 0.5$) without changing other experimental setups. The observation is consistent with the results in Figure 4. FedVote outperforms gradient quantization methods such as FedPAQ and signSGD.

## C.4 EXTENSION TO TERNARY NEURAL NETWORKS

The stochastic rounding used in the ternary neural networks, $w_i = Q_{sr}(\widetilde{w})$, is an extension of (11):

$$
w_i = \begin{cases} +1, & \text{with probability } \pi_1 = \tilde{w}_i \, \mathbb{1}(\widetilde{w}_i > 0), \\ -1, & \text{with probability } \pi_2 = -\tilde{w}_i \, \mathbb{1}(\widetilde{w}_i < 0), \\ 0, & \text{with probability } 1 - (\pi_1 + \pi_2). \end{cases} \tag{20}
$$

One can modify the normalization function to optimize neural networks with multiple quantization levels. For example, consider quaternary weight $w_i \in \{-2, -1, 1, 2\}$, the normalization function $\varphi(x)$ can be modified to $\varphi(x) = 2 \tanh(ax)$.

## C.5 BATCH NORMALIZATION IN FEDVOTE

Below we review the commonly-adopted BN function for convenience of presentation. For a one-dimensional input $x^{(j)}$ from the current batch $\mathcal{B} = \{x^{(1)}, \cdots, x^{(n_b)}\}$, the output of BN layer is formulated as

$$
y \triangleq \mathrm{BN}_{\gamma, b}(x^{(j)}) = \gamma \frac{x^{(j)} - \mu}{\sqrt{\sigma^2 + \epsilon}} + b, \tag{21}
$$

where $\gamma, b$ are learnable affine transformation parameters, and $\mu, \sigma^2$ are the mean and variance calculated over the batch samples. Note that the normal BN layer will introduce the real-valued parameters and track the statistics of the input, all of which may cause problems when being binarized in FedVote. Therefore, we choose to set the parameter-free static BN, i.e.,

$$
y' \triangleq \mathrm{BN}(x^{(j)}) = \frac{x - \mathbb{E}_{\mathcal{B}}[x^{(j)}]}{\sqrt{\mathrm{Var}_{\mathcal{B}}[x^{(j)}] + \epsilon}}. \tag{22}
$$

## D  MISSING PROOFS

### D.1  PROOF OF LEMMA 1

**Lemma 1** *(One-Shot FedVote) Let* $\mathbf{w}^* \in \mathbb{D}_2^d$ *be the optimal solution defined in (8). For the $m$th worker,* $\varepsilon_{m,i} \triangleq \mathbb{P}(w_{m,i}^{(k,\tau)} \neq w_i^*)$ *denotes the error probability of the voting result of the $i$th coordinate. Suppose the error events* $\{w_{m,i}^{(k,\tau)} \neq w_i^*\}_{m=1}^M$ *are mutually independent, and the mean error probability* $s_i = \frac{1}{M} \sum_{m=1}^M \varepsilon_{m,i}$ *is smaller than $1/2$. For the voted weight* $\mathbf{w}^{(k+1)}$, *we have*

$$
\mathbb{P}\left(w_i^{(k+1)} \neq w_i^*\right) \leqslant \left[2s_i \exp(1 - 2s_i)\right]^{\frac{M}{2}}. \tag{23}
$$

*Proof.* Let $X_{m,i} \triangleq \mathbb{1}\left(w_{m,i}^{(k,\tau)} \neq w_i^*\right)$, following Bernoulli distribution with parameter $\varepsilon_{m,i}$. Let $Y_i = \sum_{m=1}^M X_{m,i}$, we have

$$
\mathbb{P}\left(w_i^{(k+1)} \neq w_i^*\right) = \mathbb{P}\left(Y_i \geqslant \frac{M}{2}\right). \tag{24}
$$

With independent vote results from workers, $Y_i$ follows Poisson binomial distribution with mean $\mu_{Y_i} = \sum_{m=1}^{M} \varepsilon_{m,i}$. $\forall\, a > 0$, the Chernoff bound can be derived as

$$\mathbb{P}\left(Y_i \geqslant \frac{M}{2}\right) \tag{25a}$$

$$= \mathbb{P}(e^{aY_i} \geqslant e^{\frac{aM}{2}}) \tag{25b}$$

$$\overset{\text{①}}{\leqslant} \exp\left(-\frac{aM}{2}\right) \mathbb{E}\left[e^{aY_i}\right] \tag{25c}$$

$$= \exp\left(-\frac{aM}{2}\right) \prod_{m=1}^{M} \left(1 - \varepsilon_{m,i} + e^a \varepsilon_{m,i}\right) \tag{25d}$$

$$= \exp\left(-\frac{aM}{2} + \sum_{m=1}^{M} \ln\left(1 + \varepsilon_{m,i}(e^a - 1)\right)\right) \tag{25e}$$

$$\overset{\text{②}}{\leqslant} \exp\left(-\frac{aM}{2} + \sum_{m=1}^{M} \varepsilon_{m,i}(e^a - 1)\right). \tag{25f}$$

where ① is based on Markov's inequality. ② holds due to $\ln(1+x) \leqslant x$ for all $x \in (-1, \infty)$.

By assumption we have $\mu_{Y_i} < \frac{M}{2}$. Let $a = \ln\frac{M}{2\mu_{Y_i}}$, we have

$$\mathbb{P}\left(Y_i \geqslant \frac{M}{2}\right) \leqslant \frac{\exp\left(-\mu_{Y_i} + \frac{M}{2}\right)}{\left(\frac{M}{2\mu_{Y_i}}\right)^{\frac{M}{2}}} \tag{26a}$$

$$\leqslant \left(\frac{2\mu_{Y_i}}{M} \exp\left(1 - \frac{2\mu_{Y_i}}{M}\right)\right)^{\frac{M}{2}}. \tag{26b}$$

Let $s_i = \frac{\mu_{Y_i}}{M} = \frac{1}{M}\sum_{m=1}^{M} \varepsilon_{m,i}$ and substitute it into (26b), the proof is complete. $\qquad\square$

## D.2   PROOF OF LEMMA 2

**Lemma 2** *(Relationship with FedAvg) For the normalized weights, FedVote recovers FedAvg in expectation:* $\mathbb{E}\left[\widetilde{\mathbf{w}}^{(k+1)}\right] = \frac{1}{M}\sum_{m=1}^{M} \widetilde{\mathbf{w}}_m^{(k,\tau)}$, *where* $\widetilde{\mathbf{w}}^{(k+1)} = \varphi(\boldsymbol{h}^{(k+1)})$ *and* $\widetilde{\mathbf{w}}_m^{(k,\tau)} = \varphi(\boldsymbol{h}_m^{(k,\tau)})$.

*Proof.* From the inverse normalization in (14), we have $\widetilde{\mathbf{w}}^{(k+1)} = 2\boldsymbol{p}^{(k+1)} - 1$. Recall the definition of the empirical Bernoulli parameter $\boldsymbol{p}^{(k+1)}$ given by (13), we have the elementwise expectation

$$\mathbb{E}_{\boldsymbol{\pi}}\left[\widetilde{w}_i^{(k+1)}\right] = \frac{1}{M}\sum_{m=1}^{M} \left(2\widehat{\mathbb{P}}(w_{m,i}^{(k,\tau)} = 1) - 1\right) \tag{27a}$$

$$\overset{\text{①}}{=} \frac{1}{M}\sum_{m=1}^{M} \varphi(h_{m,i}^{(k,\tau)}), \tag{27b}$$

where ① follows from stochastic rounding defined in (11). Based on the definition of range normalization in (10), for local normalized weight we have $w_{m,i}^{(k,\tau)} = \varphi(h_{m,i}^{(k,\tau)})$. Substituting the result into (27b) we have

$$\mathbb{E}_{\boldsymbol{\pi}}\left[\widetilde{\mathbf{w}}^{(k+1)}\right] = \frac{1}{M}\sum_{m=1}^{M} \widetilde{\mathbf{w}}_m^{(k,\tau)}, \tag{28}$$

which completes the proof. $\qquad\square$

## D.3   PROOF OF LEMMA 3

**Lemma 3** *Suppose we have an input* $\boldsymbol{a} \in (-1, 1)^d$ *for the quantizer* $Q_{\mathrm{sr}}$ *defined in (11), then the quantization error satisfies* $\mathbb{E}\left[\left\|Q_{\mathrm{sr}}(\boldsymbol{a}) - \boldsymbol{a}\right\|_2^2 \big| \boldsymbol{a}\right] = d - \left\|\boldsymbol{a}\right\|_2^2.$

*Proof.* Let $\hat{\boldsymbol{a}} \triangleq Q_{\mathrm{sr}}(\boldsymbol{a})$, we have

$$\mathbb{E}\left[\left\|Q_{\mathrm{sr}}(\boldsymbol{a}) - \boldsymbol{a}\right\|_2^2 \Big| \boldsymbol{a}\right] = \mathbb{E}\left[\sum_{i=1}^d (\hat{a}_i - a_i)^2 \Big| \boldsymbol{a}\right] \tag{29a}$$

$$= \sum_{i=1}^d \left(\mathbb{E}\left[\hat{a}_i^2 \big| \boldsymbol{a}\right] - a_i^2\right) \tag{29b}$$

$$= \sum_{r=1}^d \left(1 - a_i^2\right) \tag{29c}$$

$$= d - \|\boldsymbol{a}\|_2^2. \tag{29d}$$

The proof is complete. $\qquad\square$

### D.4 PROOF OF LEMMA 4

**Lemma 4** *Suppose we have an input $\mathbf{x} \in \mathbb{R}^d$ for the QSGD quantizer Q. In the coarse quantization scenario, the quantization error satisfies $\mathbb{E}\left[\left\|Q(\mathbf{x}) - \mathbf{x}\right\|_2^2 \big| \mathbf{x}\right] = O\left(d^{\frac{1}{2}}\right) \|\mathbf{x}\|_2^2$.*

*Proof.* Consider a QSGD quantizer (Alistarh et al., 2017) with $s = 1$. For detailed results of other quantizers, we refer readers to Basu et al. (2020).

In particular, we have

$$Q_{\mathrm{qsgd}}\left(x_i\right) = \|\mathbf{x}\|_2 \cdot \operatorname{sgn}\left(x_i\right) \cdot \xi_i(\mathbf{x}, s), \tag{30}$$

where

$$\xi_i(\mathbf{x}, s)|_{s=1} = \begin{cases} 0 & \text{with prob. } 1 - \frac{|x_i|}{\|\mathbf{x}\|_2}, \\ 1 & \text{with prob. } \frac{|x_i|}{\|\mathbf{x}\|_2}. \end{cases} \tag{31}$$

The variance of quantization error is

$$\mathbb{E}\left[\left\|Q(\mathbf{x}) - \mathbf{x}\right\|_2^2 \big| \mathbf{x}\right] = \mathbb{E}\left[\sum_{i=1}^d \left(\hat{x}_i - x_i\right)^2 \big| \mathbf{x}\right] \tag{32a}$$

$$= \sum_{i=1}^d \left(\mathbb{E}\left[\hat{x}_i^2 \big| \mathbf{x}\right] - x_i^2\right) \tag{32b}$$

$$= \|\mathbf{x}\|_2^2 \frac{\sum_{i=1}^d |x_i|}{\|\mathbf{x}\|_2} - \|\mathbf{x}\|_2^2 \tag{32c}$$

$$= \|\mathbf{x}\|_2 \|\mathbf{x}\|_1 - \|\mathbf{x}\|_2^2 \tag{32d}$$

$$\leqslant (\sqrt{d} - 1)\|\mathbf{x}\|_2^2, \tag{32e}$$

which completes the proof. $\qquad\square$

### D.5 PROOF OF THEOREM 1

We first introduce some notations for simplicity. Let $\Delta^{(k)}$ denote the difference between two successive global latent weights, i.e.,

$$\Delta^{(k)} \triangleq \widetilde{\mathbf{w}}^{(k)} - \widetilde{\mathbf{w}}^{(k+1)}. \tag{33}$$

We use $\varepsilon_m^{(k,t)}$ to denote the stochastic gradient noise, i.e.,

$$\varepsilon^{(k,t)} \triangleq \tilde{\boldsymbol{g}}_m^{(k,t)} - \boldsymbol{g}_m^{(k,t)}. \tag{34}$$

Finally, we let $\nabla f(\widetilde{\mathbf{w}})$ denote the gradient with respect to $\widetilde{\mathbf{w}}$. The following five lemmas are presented to facilitate the proof.

**Lemma 5** *The global normalized weight $\widetilde{\mathbf{w}}^{(k)}$ can be reconstructed as the average of local binary weight, i.e.,*

$$\widetilde{\mathbf{w}}^{(k+1)} = \frac{1}{M} \sum_{m=1}^{M} \mathbf{w}_m^{(k,\tau)}. \tag{35}$$

*Proof.* See Appendix D.6. $\square$

**Lemma 6** *(Lipschitz continuity) Under Assumption 3, $\forall\, x_1,\, x_2 \in \mathbb{R}$, we have*

$$|\varphi(x_1) - \varphi(x_2)| \leqslant c_2 |x_1 - x_2|. \tag{36}$$

*Proof.* Without loss of generality, suppose $x_1 < x_2$. Based on the mean value theorem, there exists some $c \in (x_1, x_2)$ such that

$$\varphi'(c) = \frac{\varphi(x_2) - \varphi(x_1)}{x_2 - x_1}. \tag{37}$$

For the monotonically increasing function $\varphi$, we have

$$\varphi(x_2) - \varphi(x_1) = \varphi'(c)(x_2 - x_1) \tag{38a}$$

$$\overset{①}{\leqslant} c_2 (x_2 - x_1), \tag{38b}$$

where ① holds due to Assumption 3. The similar result can be obtained by assuming $x_2 < x_1$, which completes the proof. $\square$

**Lemma 7** *(Bounded weight divergence) Under Assumption 4, we have*

$$\mathbb{E}\big\|\widetilde{\mathbf{w}}_m^{(k,\tau)} - \widetilde{\mathbf{w}}^{(k)}\big\|_2^2 \leqslant (c_2 \eta)^2 \tau \left( \sum_{t=0}^{\tau-1} \mathbb{E}\big\|\boldsymbol{g}_m^{(k,t)}\big\|_2^2 + \sigma_\varepsilon^2 \right). \tag{39}$$

*Proof.* See Appendix D.7. $\square$

**Lemma 8** *Under Assumptions 2 to 5, we have*

$$\mathbb{E}\Big\langle \nabla f(\widetilde{\mathbf{w}}^{(k)}), \Delta^{(t)} \Big\rangle \geqslant \frac{c_1^2 \eta \tau}{2} \mathbb{E}\big\|\nabla f(\widetilde{\mathbf{w}}^{(k)})\big\|_2^2 \tag{40}$$

$$+ \frac{c_1^2 \eta}{4M} \big(2 - (c_2 L)^2 \eta^2 \tau(\tau - 1)\big) \sum_{m=1}^{M} \sum_{t=0}^{\tau-1} \mathbb{E}\big\|\boldsymbol{g}_m^{(k,t)}\big\|_2^2$$

$$- \frac{(c_1 c_2 L)^2 \eta^3 \tau(\tau - 1)}{4} \sigma_\varepsilon^2 - \frac{\eta(c_2^2 - c_1^2)}{M} \sum_{m=1}^{M} R_m^{(k)}, \tag{41}$$

*where $R_m^{(k)} \triangleq - \sum_{t=0}^{\tau-1} \sum_{i \notin \mathcal{I}_m^{(k,t)}} \mathbb{E}\left[ (\nabla f(\widetilde{\mathbf{w}}^{(k)}))_i (\nabla f(\widetilde{\mathbf{w}}_m^{(k,t)}))_i \right]$ and $\mathcal{I}_m^{(k,t)} \triangleq \left\{ i \mid \boldsymbol{g}_i^{(k)} \boldsymbol{g}_{m,i}^{(k,t)} \geqslant 0 \right\}$.*

*Proof.* See Appendix D.8. $\square$

**Lemma 9** *(Bounded global weight difference) Under Assumptions 2 to 5, we have*

$$\mathbb{E}\big\|\Delta^{(k)}\big\|_2^2 \leqslant \frac{(c_2 \eta)^2 \tau}{M} \sum_{m=1}^{M} \sum_{t=0}^{\tau-1} \mathbb{E}\big\|\boldsymbol{g}_m^{(k,t)}\big\|_2^2 + \frac{(c_2 \eta)^2 \tau}{M} \sigma_\varepsilon^2 + \frac{1}{M} \sigma_k^2. \tag{42}$$

*Proof.* See Appendix D.9. $\square$

**Theorem 1** *For FedVote under Assumptions 1 to 5, if the learning let the learning rate $\eta = O\left(\left(\frac{c_1}{c_2}\right)^2 \frac{1}{L\tau\sqrt{K}}\right)$, then after $K$ rounds of communication, we have*

$$\frac{1}{K}\sum_{k=0}^{K-1} c_1^2 \mathbb{E}\big\|\nabla f(\widetilde{\mathbf{w}}^{(k)})\big\|_2^2 \leqslant \frac{2\left[f(\widetilde{\mathbf{w}}^{(0)}) - f(\widetilde{\mathbf{w}}^*)\right]}{\eta\tau K} + c_2^2 L\eta \left[\frac{1}{M} + \frac{c_1^2 L\eta(\tau-1)}{2}\right]\sigma_\varepsilon^2$$

$$+ \frac{L}{\eta\tau KM}\sum_{k=0}^{K-1}\sigma_k^2 + \frac{2(c_2^2 - c_1^2)}{\tau MK}\sum_{k=0}^{K-1}\sum_{m=1}^{M} R_m^{(k)}. \tag{43}$$

*where $R_m^{(k)} \triangleq -\sum_{t=0}^{\tau-1}\sum_{i\notin\mathcal{I}_m^{(k,t)}} \mathbb{E}\left[(\nabla f(\widetilde{\mathbf{w}}^{(k)}))_i (\nabla f(\widetilde{\mathbf{w}}_m^{(k,t)}))_i\right]$ and $\mathcal{I}_m^{(k,t)} \triangleq \left\{i \mid \boldsymbol{g}_i^{(k)}\boldsymbol{g}_{m,i}^{(k,t)} \geqslant 0\right\}$.*

*Proof.* Consider the difference vector $\Delta^{(k)}$ defined in (33), we expand it as

$$\Delta^{(k)} = \widetilde{\mathbf{w}}^{(k)} - \widetilde{\mathbf{w}}^{(k+1)} \tag{44a}$$

$$\overset{①}{=} \widetilde{\mathbf{w}}^{(k)} - \frac{1}{M}\sum_{m=1}^{M}\mathbf{w}_m^{(k,\tau)} \tag{44b}$$

$$\overset{②}{=} \widetilde{\mathbf{w}}^{(k)} - \frac{1}{M}\sum_{m=1}^{M}\widetilde{\mathbf{w}}_m^{(k,\tau)} + \frac{1}{M}\sum_{m=1}^{M}\boldsymbol{\zeta}_m^{(r)}, \tag{44c}$$

where ① follows from (35), and ② holds by substituting the definition of quantization error.

From Assumption 2, we have

$$f(\widetilde{\mathbf{w}}^{(k+1)}) - f(\widetilde{\mathbf{w}}^{(k)}) \leqslant -\left\langle \nabla f(\widetilde{\mathbf{w}}^{(k)}), \Delta^{(k)}\right\rangle + \frac{L}{2}\big\|\Delta^{(k)}\big\|_2^2. \tag{45a}$$

Let $\frac{(L\eta)^2\tau(\tau-1)}{2} + \frac{L\eta\tau}{c_1^2} \leqslant \frac{1}{c_2^2}$, take the expectation on both sides, and use Lemmas 8–9:

$$\mathbb{E}\left[f(\widetilde{\mathbf{w}}^{(k+1)}) - f(\widetilde{\mathbf{w}}^{(k)})\right] \leqslant -\frac{c_1^2\eta\tau}{2}\mathbb{E}\big\|\nabla f(\widetilde{\mathbf{w}}^{(k)})\big\|_2^2$$

$$- \frac{(c_1 c_2)^2\eta}{4M}\left(\frac{2}{c_2^2} - L^2\eta^2\tau(\tau-1) - \frac{2L\eta\tau}{c_1^2}\right)\sum_{m=1}^{M}\sum_{t=0}^{\tau-1}\mathbb{E}\big\|\boldsymbol{g}_m^{(k,t)}\big\|_2^2$$

$$+ \frac{(c_2\eta)^2 L\tau}{4}\left(\frac{2}{M} + c_1^2 L\eta(\tau-1)\right)\sigma_\varepsilon^2 + \frac{\eta(c_2^2 - c_1^2)}{M}\sum_{m=1}^{M} R_m^{(k)} + \frac{L}{2M}\sigma_k^2 \tag{46a}$$

$$\overset{①}{\leqslant} -\frac{c_1^2\eta\tau}{2}\mathbb{E}\big\|\nabla f(\widetilde{\mathbf{w}}^{(k)})\big\|_2^2 + \frac{(c_2\eta)^2 L\tau}{4}\left(\frac{2}{M} + c_1^2 L\eta(\tau-1)\right)\sigma_\varepsilon^2$$

$$+ \frac{\eta(c_2^2 - c_1^2)}{M}\sum_{m=1}^{M} R_m^{(k)} + \frac{L}{2M}\sigma_k^2, \tag{46b}$$

where ① follows from the restrictions on learning rate. We rewrite the restriction as

$$\eta \leqslant \frac{-\frac{L\tau}{c_1^2} + \sqrt{\frac{L^2\tau^2}{c_1^4} + \frac{2L^2\tau(\tau-1)}{c_2^2}}}{L^2\tau(\tau-1)} \tag{47a}$$

$$\overset{①}{\leqslant} \frac{-1 + \sqrt{1 + \frac{2c_1^4}{c_2^2}}}{L(\tau-1)c_1^2} \tag{47b}$$

$$\overset{②}{\leqslant} \frac{c_1^2}{L(\tau-1)c_2^2}, \tag{47c}$$

where in ① we use $\tau(\tau-1) \leqslant \tau^2$, and ② holds due to the Bernoulli inequality $(1+x)^{\frac{1}{2}} \leqslant 1 + \frac{1}{2}x, \ \forall\, x \in [-1, \infty)$. Based on (47c), we set the learning rate $\eta = O\left(\left(\frac{c_1}{c_2}\right)^2 \frac{1}{L\tau\sqrt{K}}\right)$. Summing

up over $K$ communication rounds yields

$$\frac{1}{K}\sum_{k=0}^{K-1}c_1^2\mathbb{E}\big\|\nabla f(\widetilde{\mathbf{w}}^{(k)})\big\|_2^2 \leqslant \frac{2\left[f(\widetilde{\mathbf{w}}^{(0)}) - f(\widetilde{\mathbf{w}}^*)\right]}{\eta\tau K} + c_2^2 L\eta\left[\frac{1}{M} + \frac{c_1^2 L\eta(\tau-1)}{2}\right]\sigma_\varepsilon^2$$

$$+ \frac{L}{\eta\tau KM}\sum_{k=0}^{K-1}\sigma_k^2 + \frac{2(c_2^2 - c_1^2)}{\tau MK}\sum_{k=0}^{K-1}\sum_{m=1}^{M}R_m^{(k)}. \tag{48a}$$

$\square$

### D.6 PROOF OF LEMMA 5

*Proof.* From the reconstruction rule (14), we have

$$\widetilde{\mathbf{w}}^{(k+1)} = 2\boldsymbol{p}^{(k+1)} - 1. \tag{49}$$

The $i$th entry of $\boldsymbol{p}^{(k)}$ is defined in (13) as:

$$p_i^{(k+1)} = \frac{1}{M}\sum_{m=1}^{M}\mathbb{1}\left(w_{m,i}^{(k,\tau)} = 1\right). \tag{50}$$

Substituting (50) into (49) yields

$$\widetilde{w}_i^{(k+1)} = \frac{1}{M}\sum_{m=1}^{M}\left[2\mathbb{1}\left(w_{m,i}^{(k,\tau)} = 1\right) - 1\right]. \tag{51}$$

Note that

$$2\mathbb{1}\left(w_{m,i}^{(k,\tau)} = 1\right) - 1 = \begin{cases} +1, & w_m^{(k,\tau)} = +1, \\ -1, & w_m^{(k,\tau)} = -1, \end{cases} \tag{52}$$

we have

$$\widetilde{w}_i^{(k+1)} = \frac{1}{M}\sum_{m=1}^{M}w_i^{(k)}, \tag{53}$$

which completes the proof. $\square$

### D.7 PROOF OF LEMMA 7

*Proof.* With the local initialization and update method described in Algorithm 1, we have

$$\mathbb{E}\big\|\widetilde{\mathbf{w}}_m^{(k,\tau)} - \widetilde{\mathbf{w}}^{(k)}\big\|_2^2 = \mathbb{E}\big\|\varphi(\boldsymbol{h}_m^{(k,\tau)}) - \varphi(\boldsymbol{h}_m^{(k,0)})\big\|_2^2 \tag{54a}$$

$$\overset{①}{\leqslant} c_2^2\mathbb{E}\big\|\boldsymbol{h}_m^{(k,\tau)} - \boldsymbol{h}_m^{(k,0)}\big\|_2^2 \tag{54b}$$

$$= c_2^2\mathbb{E}\bigg\|\sum_{t=0}^{\tau-1}-\eta\,\tilde{\boldsymbol{g}}_m^{(k,t)}\bigg\|_2^2 \tag{54c}$$

$$= (c_2\eta)^2\mathbb{E}\bigg\|\sum_{t=0}^{\tau-1}\left(\boldsymbol{g}_m^{(k,t)} + \boldsymbol{\varepsilon}_m^{(k,t)}\right)\bigg\|_2^2 \tag{54d}$$

$$= (c_2\eta)^2\bigg(\mathbb{E}\bigg\|\sum_{t=0}^{\tau-1}\boldsymbol{g}_m^{(k,t)}\bigg\|_2^2 + \mathbb{E}\bigg\|\sum_{t=0}^{\tau-1}\boldsymbol{\varepsilon}_m^{(k,t)}\bigg\|_2^2\bigg) \tag{54e}$$

$$\leqslant (c_2\eta)^2\tau\bigg(\sum_{t=0}^{\tau-1}\mathbb{E}\big\|\boldsymbol{g}_m^{(k,t)}\big\|_2^2 + \sigma_\varepsilon^2\bigg), \tag{54f}$$

where ① comes from the Lipschitz condition in Lemma 6. $\square$

### D.8 PROOF OF LEMMA 8

*Proof.* We have

$$\mathbb{E}\left\langle \nabla f(\widetilde{\mathbf{w}}^{(k)}), \Delta^{(k)} \right\rangle$$

$$= \mathbb{E}\left\langle \nabla f(\widetilde{\mathbf{w}}^{(k)}), \widetilde{\mathbf{w}}^{(k)} - \frac{1}{M}\sum_{m=1}^{M}\widetilde{\mathbf{w}}_m^{(k,\tau)} + \frac{1}{M}\sum_{m=1}^{M}\zeta_m^{(k)} \right\rangle \tag{55a}$$

$$\overset{\text{①}}{=} \mathbb{E}\left\langle \nabla f(\widetilde{\mathbf{w}}^{(k)}), \widetilde{\mathbf{w}}^{(k)} - \frac{1}{M}\sum_{m=1}^{M}\widetilde{\mathbf{w}}_m^{(k,\tau)} \right\rangle \tag{55b}$$

$$= \mathbb{E}\left\langle \nabla f(\widetilde{\mathbf{w}}^{(k)}), \frac{1}{M}\sum_{m=1}^{M}\varphi(\boldsymbol{h}^{(k)}) - \varphi(\boldsymbol{h}_m^{(k,\tau)}) \right\rangle, \tag{55c}$$

where ① follows from Assumption 5. Based on the mean value theorem, for $h_i^{(k)}, h_{m,i}^{(t,\tau)} \in \mathbb{R}$, there exists some $c_{m,i}^{(k)} \in \mathbb{H}_{m,i}^{(r)}$ such that

$$\varphi'(c_{m,i}^{(k)}) = \frac{\varphi(h_i^{(k)}) - \varphi(h_{m,i}^{(k,\tau)})}{h_i^{(k)} - h_{m,i}^{(k,\tau)}}, \tag{56}$$

where $\mathbb{H}_{m,i}^{(r)}$ is an open interval with endpoints $h_i^{(k)}$ and $h_{m,i}^{(k,\tau)}$:

$$\mathbb{H}_{m,i}^{(r)} = \begin{cases} (h_i^{(k)}, h_{m,i}^{(k,\tau)}), & \text{if } h_i^{(k)} < h_{m,i}^{(k,\tau)}, \\ (h_{m,i}^{(k,\tau)}, h_i^{(k)}), & \text{otherwise.} \end{cases} \tag{57}$$

Let $C_m^{(k)} = \text{diag}\left(\varphi'(c_{m,1}^{(k)}), \ldots, \varphi'(c_{m,d}^{(k)})\right)$, we have

$$\varphi(\boldsymbol{h}^{(k)}) - \varphi(\boldsymbol{h}_m^{(k,\tau)}) = C_m^{(k)}(\boldsymbol{h}^{(k)} - \boldsymbol{h}_m^{(k,\tau)}) = \eta\, C_m^{(k)}\sum_{t=0}^{\tau-1}\left(\boldsymbol{g}_m^{(k,t)} + \boldsymbol{\varepsilon}_m^{(k,t)}\right), \tag{58}$$

Substituting (58) into (55c) yields

$$\mathbb{E}\left\langle \nabla f(\widetilde{\mathbf{w}}^{(k)}), \Delta^{(k)} \right\rangle = \frac{\eta}{M}\sum_{m=1}^{M}\sum_{t=0}^{\tau-1}\mathbb{E}\left\langle \nabla f(\widetilde{\mathbf{w}}^{(k)}), C_m^{(k)}\boldsymbol{g}_m^{(k,t)} \right\rangle. \tag{59}$$

According to the chain rule, $\boldsymbol{g}_m^{(k,t)}$ can be written as

$$\boldsymbol{g}_m^{(k,t)} = \nabla_{\boldsymbol{h}} f_m(\varphi(\boldsymbol{h}_m^{(k,t)})) = D_m^{(k,t)}\nabla_{\widetilde{\mathbf{w}}} f_m(\widetilde{\mathbf{w}}_m^{(k,t)}), \tag{60}$$

where $D_m^{(k,t)} = \text{diag}\left(\frac{\mathrm{d}\varphi}{\mathrm{d}h_{m,1}^{(k,t)}}, \ldots, \frac{\mathrm{d}\varphi}{\mathrm{d}h_{m,d}^{(k,t)}}\right)$. To simplify the notations, let $B_m^{(k,t)} = C_m^{(k)}D_m^{(k,t)}$. Note that $B_m^{(k,t)}$ is still a diagonal matrix, where the $i$th diagonal element $b_{m,i}^{(k,t)}$ is

$$b_{m,i}^{(k,t)} \triangleq (B_m^{(k,t)})_{i,i} = \frac{\mathrm{d}\varphi}{\mathrm{d}c_{m,i}^{(k,\tau)}}\frac{\mathrm{d}\varphi}{\mathrm{d}h_{m,i}^{(k,t)}}. \tag{61}$$

Substituting (60) into (59) we have

$$\mathbb{E}\left\langle \nabla f(\widetilde{\mathbf{w}}^{(k)}), \Delta^{(k)} \right\rangle = \frac{\eta}{M}\sum_{m=1}^{M}\sum_{t=0}^{\tau-1}\mathbb{E}\left\langle \nabla f(\widetilde{\mathbf{w}}^{(k)}), B_m^{(k,t)}\nabla f_m(\widetilde{\mathbf{w}}_m^{(k,t)}) \right\rangle. \tag{62}$$

We first focus on the following inner product:

$$\left\langle \nabla f(\widetilde{\mathbf{w}}^{(k)}), B_m^{(k,t)}\nabla f_m(\widetilde{\mathbf{w}}_m^{(k,t)}) \right\rangle = \sum_{i=1}^{d}(\nabla f(\widetilde{\mathbf{w}}^{(k)}))_i \times b_{m,i}^{(k,t)}(\nabla f_m(\widetilde{\mathbf{w}}_m^{(k,t)}))_i, \tag{63}$$

where $(\nabla f)_i$ denotes the $i$th entry of the gradient vector. Consider an index set $\mathcal{I}_m^{(k,t)}$ defined as

$$\mathcal{I}_m^{(k,t)} \triangleq \left\{ i \in \{1,\dots,d\} \mid (\nabla f(\widetilde{\mathbf{w}}^{(k)}))_i (\nabla f_m(\widetilde{\mathbf{w}}_m^{(k,t)}))_i \geqslant 0 \right\}. \tag{64}$$

Since the sign of $(\nabla f(\widetilde{\mathbf{w}}^{(k)}))_i (\nabla f_m(\widetilde{\mathbf{w}}_m^{(k,t)}))_i$ is equal to $\boldsymbol{g}_i^{(k)} \boldsymbol{g}_{m,i}^{(k,t)}$, the index set $\mathcal{I}_m^{(k,t)}$ can also be written as

$$\mathcal{I}_m^{(k,t)} \triangleq \left\{ i \in \{1,\dots,d\} \mid \boldsymbol{g}_i^{(k)} \boldsymbol{g}_{m,i}^{(k,t)} \geqslant 0 \right\}. \tag{65}$$

The result in (63) can be bounded as

$$\mathbb{E} \left\langle \nabla f(\widetilde{\mathbf{w}}^{(k)}), B_m^{(k,t)} \nabla f_m(\widetilde{\mathbf{w}}_m^{(k,t)}) \right\rangle \tag{66a}$$

$$\overset{\textcircled{1}}{\geqslant} c_1^2 \sum_{i \in \mathcal{I}_m^{(k,t)}} \mathbb{E} \left[ (\nabla f(\widetilde{\mathbf{w}}^{(k)}))_i (\nabla f_m(\widetilde{\mathbf{w}}_m^{(k,t)}))_i \right] + c_2^2 \sum_{i \notin \mathcal{I}_m^{(k,t)}} \mathbb{E} \left[ (\nabla f(\widetilde{\mathbf{w}}^{(k)}))_i (\nabla f_m(\widetilde{\mathbf{w}}_m^{(k,t)}))_i \right]$$
$$\tag{66b}$$

$$= c_1^2 \mathbb{E} \left\langle \nabla f(\widetilde{\mathbf{w}}^{(k)}), \nabla f_m(\widetilde{\mathbf{w}}_m^{(k,t)}) \right\rangle + (c_2^2 - c_1^2) \sum_{i \notin \mathcal{I}_m^{(k,t)}} \mathbb{E} \left[ (\nabla f(\widetilde{\mathbf{w}}^{(k)}))_i (\nabla f_m(\widetilde{\mathbf{w}}_m^{(k,t)}))_i \right], \tag{66c}$$

where $\textcircled{1}$ follows from Assumption 3. To simplify the notation, let $R_m^{(k)}$ denote the accumulative gradient divergence, namely,

$$R_m^{(k)} \triangleq -\sum_{t=0}^{\tau-1} \sum_{i \notin \mathcal{I}_m^{(k,t)}} \mathbb{E} \left[ (\nabla f(\widetilde{\mathbf{w}}^{(k)}))_i (\nabla f(\widetilde{\mathbf{w}}_m^{(k,t)}))_i \right]. \tag{67}$$

The expected inner product in (62) can be bounded as

$$\mathbb{E} \left\langle \nabla f(\widetilde{\mathbf{w}}^{(k)}), \Delta^{(k)} \right\rangle$$

$$\geqslant \frac{c_1^2 \eta}{M} \sum_{m=1}^{M} \sum_{t=0}^{\tau-1} \mathbb{E} \left\langle \nabla f(\widetilde{\mathbf{w}}^{(k)}), \nabla f_m(\widetilde{\mathbf{w}}_m^{(k,t)}) \right\rangle - \frac{\eta(c_2^2 - c_1^2)}{M} \sum_{m=1}^{M} R_m^{(k)} \tag{68a}$$

$$\overset{\textcircled{1}}{=} \frac{c_1^2 \eta}{2M} \sum_{m=1}^{M} \sum_{t=0}^{\tau-1} \left( \mathbb{E} \left\| \nabla f(\widetilde{\mathbf{w}}^{(k)}) \right\|_2^2 + \mathbb{E} \left\| \nabla f(\widetilde{\mathbf{w}}_m^{(k,t)}) \right\|_2^2 \right.$$

$$\left. - \mathbb{E} \left\| \nabla f(\widetilde{\mathbf{w}}_m^{(k,t)}) - \nabla f(\widetilde{\mathbf{w}}^{(k)}) \right\|_2^2 \right) - \frac{\eta(c_2^2 - c_1^2)}{M} \sum_{m=1}^{M} R_m^{(k)} \tag{68b}$$

$$\overset{\textcircled{2}}{\geqslant} \frac{c_1^2 \eta \tau}{2} \left\| \nabla f(\widetilde{\mathbf{w}}^{(k)}) \right\|_2^2 + \frac{c_1^2 \eta}{2M} \sum_{m=1}^{M} \sum_{t=0}^{\tau-1} \left( \mathbb{E} \left\| \nabla f(\widetilde{\mathbf{w}}_m^{(k,t)}) \right\|_2^2 \right.$$

$$\left. - L^2 \mathbb{E} \left\| \widetilde{\mathbf{w}}_m^{(k,t)} - \widetilde{\mathbf{w}}^{(k)} \right\|_2^2 \right) - \frac{\eta(c_2^2 - c_1^2)}{M} \sum_{m=1}^{M} R_m^{(k)}, \tag{68c}$$

where $\textcircled{1}$ follows from $2\langle \boldsymbol{a}, \boldsymbol{b} \rangle = \|\boldsymbol{a}\|_2^2 + \|\boldsymbol{b}\|_2^2 - \|\boldsymbol{a} - \boldsymbol{b}\|_2^2$, and $\textcircled{2}$ holds due to Assumption 2. From Lemma 7, we can show that

$$\mathbb{E} \left\| \widetilde{\mathbf{w}}_m^{(k,t)} - \widetilde{\mathbf{w}}^{(k)} \right\|_2^2 \leqslant (c_2 \eta)^2 t \left( \sum_{n=0}^{t-1} \left\| \boldsymbol{g}_m^{(k,n)} \right\|_2^2 + \sigma_\varepsilon^2 \right), \tag{69}$$

where $t = 1, \ldots, \tau - 1$. Substituting (69) in (68c) yields

$$
\mathbb{E}\left\langle \nabla f(\widetilde{\mathbf{w}}^{(k)}), \Delta^{(k)} \right\rangle
$$

$$
\geqslant \frac{c_1^2 \eta \tau}{2} \mathbb{E}\big\|\nabla f(\widetilde{\mathbf{w}}^{(k)})\big\|_2^2 + \frac{c_1^2 \eta}{2M} \sum_{m=1}^{M} \sum_{t=0}^{\tau-1} \mathbb{E}\big\|\nabla f(\widetilde{\mathbf{w}}_m^{(k,t)})\big\|_2^2
$$

$$
- \frac{(c_1 c_2 L)^2 \eta^3}{4M} \sum_{m=1}^{M} \sum_{n=0}^{\tau-1} (\tau(\tau-1) - n(n+1)) \, \mathbb{E}\big\|\boldsymbol{g}_m^{(k,n)}\big\|_2^2
$$

$$
- \frac{(c_1 c_2 L)^2 \eta^3 \tau(\tau-1)}{4} \sigma_\varepsilon^2 - \frac{\eta(c_2^2 - c_1^2)}{M} \sum_{m=1}^{M} R_m^{(k)} \tag{70a}
$$

$$
\overset{\text{①}}{\geqslant} \frac{c_1^2 \eta \tau}{2} \mathbb{E}\big\|\nabla f(\widetilde{\mathbf{w}}^{(k)})\big\|_2^2 + \frac{c_1^2 \eta}{2M} \sum_{m=1}^{M} \sum_{t=0}^{\tau-1} \mathbb{E}\big\|\nabla f(\widetilde{\mathbf{w}}_m^{(k,t)})\big\|_2^2
$$

$$
- \frac{\eta}{4M} (c_1 c_2 L)^2 \eta^2 \tau(\tau-1) \sum_{m=1}^{M} \sum_{t=0}^{\tau-1} \mathbb{E}\big\|\boldsymbol{g}_m^{(k,t)}\big\|_2^2
$$

$$
- \frac{(c_1 c_2 L)^2 \eta^3 \tau(\tau-1)}{4} \sigma_\varepsilon^2 - \frac{\eta(c_2^2 - c_1^2)}{M} \sum_{m=1}^{M} R_m^{(k)} \tag{70b}
$$

$$
\overset{\text{②}}{\geqslant} \frac{c_1^2 \eta \tau}{2} \mathbb{E}\big\|\nabla f(\widetilde{\mathbf{w}}^{(k)})\big\|_2^2 + \frac{c_1^2 \eta}{4M} \left(2 - (c_2 L)^2 \eta^2 \tau(\tau-1)\right) \sum_{m=1}^{M} \sum_{t=0}^{\tau-1} \mathbb{E}\big\|\boldsymbol{g}_m^{(k,t)}\big\|_2^2
$$

$$
- \frac{(c_1 c_2 L)^2 \eta^3 \tau(\tau-1)}{4} \sigma_\varepsilon^2 - \frac{\eta(c_2^2 - c_1^2)}{M} \sum_{m=1}^{M} R_m^{(k)} \tag{70c}
$$

where ① follows from $\tau(\tau-1) - n(n+1) \leqslant \tau(\tau-1)$, and ② holds due to the chain rule in (60) and Assumption 3.

$\square$

## D.9    PROOF OF LEMMA 9

*Proof.* We have

$$
\mathbb{E}\big\|\Delta^{(k)}\big\|_2^2 = \mathbb{E}\left\| \widetilde{\mathbf{w}}^{(k)} - \frac{1}{M} \sum_{m=1}^{M} \widetilde{\mathbf{w}}_m^{(k,\tau)} + \frac{1}{M} \sum_{m=1}^{M} \zeta_m^{(k)} \right\|_2^2 \tag{71a}
$$

$$
= \mathbb{E}\left\| \frac{1}{M} \sum_{m=1}^{M} \varphi(\boldsymbol{h}^{(k)}) - \varphi(\boldsymbol{h}_m^{(k,\tau)}) + \frac{1}{M} \sum_{m=1}^{M} \zeta_m^{(k)} \right\|_2^2 \tag{71b}
$$

$$
= \mathbb{E}\left\| \frac{1}{M} \sum_{m=1}^{M} \varphi(\boldsymbol{h}^{(k)}) - \varphi(\boldsymbol{h}_m^{(k,\tau)}) \right\|_2^2 + \mathbb{E}\left\| \frac{1}{M} \sum_{m=1}^{M} \zeta_m^{(k)} \right\|_2^2
$$

$$
+ 2\mathbb{E}\left\langle \frac{1}{M} \sum_{m=1}^{M} \varphi(\boldsymbol{h}^{(k)}) - \varphi(\boldsymbol{h}_m^{(k,\tau)}), \frac{1}{M} \sum_{m=1}^{M} \zeta_m^{(t)} \right\rangle \tag{71c}
$$

$$
\overset{\text{①}}{=} \mathbb{E}\left\| \frac{1}{M} \sum_{m=1}^{M} \varphi(\boldsymbol{h}^{(k)}) - \varphi(\boldsymbol{h}_m^{(k,\tau)}) \right\|_2^2 + \mathbb{E}\left\| \frac{1}{M} \sum_{m=1}^{M} \zeta_m^{(k)} \right\|_2^2, \tag{71d}
$$

where ① comes from Assumption 5. For the first term in (71d) we have

$$\mathbb{E}\left\|\frac{1}{M}\sum_{m=1}^{M}(\varphi(\boldsymbol{h}^{(k)}) - \varphi(\boldsymbol{h}_m^{(k,\tau)}))\right\|_2^2$$

$$\leqslant \frac{1}{M}\sum_{m=1}^{M}\mathbb{E}\left\|\varphi(\boldsymbol{h}^{(k)}) - \varphi(\boldsymbol{h}_m^{(k,\tau)})\right\|_2^2 \tag{72a}$$

$$\overset{①}{\leqslant} \frac{c_2^2}{M}\sum_{m=1}^{M}\left\|\boldsymbol{h}^{(k)} - \boldsymbol{h}_m^{(k,\tau)}\right\|_2^2 \tag{72b}$$

$$\overset{②}{=} \frac{(c_2\eta)^2}{M}\sum_{m=1}^{M}\left\|-\eta\sum_{t=0}^{\tau-1}\tilde{\boldsymbol{g}}_m^{(k,t)}\right\|_2^2 \tag{72c}$$

$$= \frac{(c_2\eta)^2}{M}\sum_{m=1}^{M}\left\|\sum_{t=0}^{\tau-1}(\boldsymbol{g}_m^{(k,t)} + \boldsymbol{\varepsilon}_m^{(k,t)})\right\|_2^2 \tag{72d}$$

$$= \frac{(c_2\eta)^2\tau}{M}\sum_{m=1}^{M}\sum_{t=0}^{\tau-1}\mathbb{E}\left\|\boldsymbol{g}_m^{(k,t)}\right\|_2^2 + \frac{(c_2\eta)^2\tau}{M}\sigma_\varepsilon^2, \tag{72e}$$

where ① is due to Lemma 6 and ② comes from (9). For the second term in (71d) we have

$$\mathbb{E}\left\|\frac{1}{M}\sum_{m=1}^{M}\boldsymbol{\zeta}_m^{(k)}\right\|_2^2 \leqslant \frac{1}{M}\sigma_k^2. \tag{73}$$

Combing the results of (72e) and (73) completes the proof.

$\square$

