# OpenReview forum: "Federated Learning via Plurality Vote"
_ICLR.cc/2022/Conference — ICLR 2022 Submitted_

### Official Review · Reviewer_Cpmm · 2021-11-02

**Correctness:** 3
**Technical Novelty And Significance:** 3
**Empirical Novelty And Significance:** 2
**Recommendation:** 5
**Confidence:** 3

**Main Review:**

**Pros**:
- The paper is well-written and related works are well-address to my knowledge.
- Has a nice introduction on quantized neural networks, which is good for readers that are not familiar with the topic.

**My Questions**: I have some questions on the theory side of the paper.

- The author(s) described the convergence of FedVote in terms of $|| f(\tilde{w}^{(k)}) ||^2$, which is a valid measure of stationarity for continuous problems. I wonder what is the conclusion to the discrete problem? For example, we have a stationary point $w^*$ such that $|| f( \tilde{w}^* ) || = 0$. Then what conclusion can we obtain for the following problem
$$
	\min_{w \in \mathbb{D}^2_n} f(w).
$$
Can we say that we are at a global minimum in the convex case?

- In Lemma 1, the author(s) define $s_i =\frac{2}{M} \sum_{i=1}^M \epsilon_{m,  i}$. From the definition we know $s_i \in (0,  2)$. Why the author(s) conclude that $s_i \in (0, 1)$?

-  Theorem 1 describes the convergence of FedVote for $\eta$ that satisfies the assumption. I wonder why the author(s) do not give a Corollary on the convergence rate for a particular choice of $\eta$ (which is usually done in the literature, e.g., see the analysis in [1])? It is not straightforward for readers to see the convergence (or non-convergence) without a concrete choice of $\eta$.

- In Remark 2, the author(s) mention that the variance of quantization error will impede the convergence, which also holds for FedPAQ. However, I do not see any non-convergence issue of FedPAQ from [Theorem 2, 1]. Could the author(s) explain more on the non-convergence of FedPAQ?

- On the theory side, could the author(s) elaborate the advantages of the proposed method over FedPAQ?

[1] Reisizadeh et al. FedPAQ: A Communication-Efficient Federated Learning Method with Periodic Averaging and Quantization. AISTATS 2020.

**Summary Of The Paper:**

The author(s) proposed a new algorithm based on plurality voting for federated learning with quantized gradients. Convergence theory is developed and experiments on both iid and non-iid datasets are conducted to evaluate the proposed method.

**Summary Of The Review:**

I have some questions about the theoretical analysis of this paper. I hope that the author(s) could kindly elaborate.

---

> ### Author Response · Authors · 2021-11-20
> **Response to Reviewer Cpmm**
>
> 1. We thank the reviewer for raising this question.
> If we have $\left\\|\nabla f\left(\widetilde{\mathbf{w}}^\star\right)\right\\|=0$, we cannot say that the quantized version $\mathbf{w}$ is the global minimum of $\min_{\mathbf{w} \in \mathbb{D}_n^2} f(\mathbf{w})$.
> However, the difference between the normalized weight $\widetilde{\mathbf{w}}^\star$ and the binary weight $\mathbf{w}$ is small, the difference between neural network predictions $f(\widetilde{\mathbf{w}}^\star)$ and $f(\mathbf{w})$ is also small; we conclude that $\mathbf{w}$ is a good approximation of the global minimum of the discrete problem.
>
> 2. We restrict the averaged error probability across all workers, $\frac{1}{M} \sum_{m=1}^m \epsilon_{m,i}$, to be upper bounded by $1/2$.
> The latter assumption for the overall error is the least stringent for the upper bound of Lemma~1 to hold, and is empirically justified by the fact that FedVote learning algorithm converges in practice.
> We have modified Lemma 1 to avoid confusion:
> - Suppose the error events $\\{w^{(k, \tau)}\_{m, i} \neq w\_{i}^{*}\\}\_{m=1}^M$  are mutually independent, and the mean error probability $s_i = \frac{1}{M}\sum_{m=1}^M \varepsilon_{m,i}$ is smaller than $\frac{1}{2}$.
>
> 3. We thank the reviewer for the suggestion. We modified the proof in Appendix D.5 to provide more details on the range of $\eta$ and revised Theorem 1 in Section 5 accordingly:
> - ... let the learning rate $\eta = O\left((\frac{c_1}{c_2})^2 \frac{1}{L\tau \sqrt{K}}\right)$, then ...
>
> 4. > Could the author(s) explain more on the non-convergence of FedPAQ?
>
>     We thank the reviewer for raising this question.
> In [1], the quantization error is bounded by employing the signal to quantization noise ratio (SQNR), i.e.,  $\mathbb{E}\left[\|Q(\mathbf{x})-\mathbf{x}\|^{2} \mid \mathbf{x}\right] \leq q\|\mathbf{x}\|^{2}$.
> In [Theorem 2, 1], the second term in the upper bound $N_{1} \frac{1}{\sqrt{T}}$, $N_1 = (1+q) \frac{\sigma^{2}}{n}\left(1+\frac{n(n-r)}{r(n-1)}\right)$, is related to the coefficient $q$;
> the third term contains $\tau$, which is implicitly related to the coefficient $q$.
> In Lemma 4， we discussed that in the coarse quantization scenario, $q = O(d^{1/2})$, where $d$ is the dimension of the input vector.
> For a model weight with a large input dimension $d$, such as the fully-connected layers in the VGG networks, the quantization will impede the convergence.
>
> 5. > could the author(s) elaborate the advantages of the proposed method over FedPAQ?
>
>     In our work, instead of using the signal to quantization noise ratio (SQNR), we model the quantization error as a function of the input dimension $d$.
> We have added the sentence in Remark 2 of Section 5:
> ``Following the analysis framework in Theorem 1, the order-wise convergence rate is $\frac{1}{K} \sum_{k=0}^{K-1} \mathbb{E}\left\|\nabla f\left(\mathbf{w}^{(k)}\right)\right\|_{2}^{2}=\mathcal{O}(1/\sqrt{K}) + E(d)$, where $E(d)$ is the error introduced by the quantization.''
> \\[10pt]
> In Remark 2, we have clarified that $E(d) = O(d)$ for our scheme, while $E(d) = O(d^{3/2})$ for FedPAQ, which shows the advantages in terms of the quantization choice.

---

> > ### Comment · Reviewer_Cpmm · 2021-11-28
> > **Thanks for the clarification**
> >
> > I would like to thank the author(s) for answering my questions.
> >
> > - For Q1, I am not very satisfied with the answer. I do not find any theoretical characterization of the distance between the normalized weight $\tilde{w}^*$ and the binary weight $w$ in the paper. The conclusion from Theorem 1 is still kind of vague to me, it gives a convergence for the continuous problem (eq. (2)) and it is not clear to me how can we quantify how well the solution approximate to the discrete problem (eq. (8)).
> >
> > - For Q4. In [Theorem 2, 1], the term $N_1$ and $N_2$ are not related to $K$, so the best solution from FedPAQ will converge to a stationary point as $K$ goes to infinity. But the convergence you obtained as a term $E(d)$ and it may not converge to a stationary point even as $K$ goes to infinity. Seems that this is a weak point compared with FedPAQ. Also, I do not find the definition of $E(d)$.
> >
> > I am not completely satisfied with the revision and I am keeping my score unchanged.

---

### Official Review · Reviewer_B8sD · 2021-11-03

**Correctness:** 3
**Technical Novelty And Significance:** 2
**Empirical Novelty And Significance:** 2
**Recommendation:** 3
**Confidence:** 3

**Main Review:**

I like the topic of this paper, but I have some concerns as below.

1. Explanation on the huge gap between FedVote and SignSGD with majority vote [Bernstein’18].
    1. It seems like the basic concept of FedVote is from [Bernstein’18]. I’m bit curious why FedVote has some gap compared with [Bernstein’18]. Is it because of the weighted voting strategy of FedVote? Can we do some ablation study to check what made FedVote such powerful, compared with [Bernstein’18]?
2. Comparison with other Byzantine-tolerant schemes
    1. There have been many previous works on Byzantine-tolerant schemes, e.g., using coding scheme to defend Byzantine attacks, DRACO[Chen’18], DETOX[Rajput’19], Election coding [Sohn’20], SignGuard[Xu’21], ByzShield [Konstantinidis’21]. Especially, the last four works focus on the voting-based model/gradient averaging. I think this paper should at least mention these works, and also compare the performance.
    2. There have been naive methods of using median or mean of median schemes for tolerating Byzantines, starting from 2017, e.g., KRUM, multi-KRUM. I guess this paper didn’t compare with these naive methods also.
    3. After the [Bernstein’18], there have been upcoming works [Karimireddy’19] suggesting that error feedback fixes the issue of SignSGD. I’m not sure whether the authors compared their work with this scheme.
    4. So in general, I thought that the literature search and comparison is bit weak. SignSGD is 3-year old paper, and there have been many other papers using voting-based aggregation for Byzantine tolerance.

[Bernstein’18] https://arxiv.org/abs/1802.04434
[Chen’18] https://arxiv.org/abs/1803.09877
[Karimireddy’19] http://proceedings.mlr.press/v97/karimireddy19a/karimireddy19a.pdf
[Rajput’19] https://proceedings.neurips.cc/paper/2019/hash/415185ea244ea2b2bedeb0449b926802-Abstract.html
[Sohn’20] https://proceedings.neurips.cc/paper/2020/hash/a7f0d2b95c60161b3f3c82f764b1d1c9-Abstract.html
[Xu’21] https://arxiv.org/pdf/2109.05872.pdf
[Konstantinidis’21] https://arxiv.org/abs/2010.04902

**Summary Of The Paper:**

This paper provides a voting-based quantized model averaging method for federated learning. The convergence is proved and the experimental results on Fashion-MNIST and CIFAR-10 show that the proposed scheme outperforms existing schemes.


**Summary Of The Review:**

This is an interesting paper, but the idea is quite similar to SignSGD paper, and has no thorough comparison with existing works. Better comparison & ablation are needed to better understand why FedVote works better than existing schemes.

---

> ### Author Response · Authors · 2021-11-20
> **Response to Reviewer B8sD**
>
> 1. We thank the reviewer for raising this question.
>     > Is it because of the weighted voting strategy of FedVote?
>
>     It is not because of the weighted voting. The advantage of FedVote over signSGD [Bernstein’18] comes from the local update process that signSGD does not have. Communication is a bottleneck in federated learning, and local update may be adopted to accelerate convergence (Reisizadeh et al., 2020). In signSGD, the local update strategy is not helpful hence not used because the magnitude of the gradient is discarded. If the local update strategy is not used in signSGD, in each communication round, information coming from the sign of the gradient is limited. In contrast, FedVote inherently optimizes a binary neural network and takes the advantage of client local update to accelerate the convergence. In each communication round, each bit indicates the choice of model weight and contains more information.
>
>     > Can we do some ablation study to check what made FedVote such powerful, compared with [Bernstein’18]?
>
>     As we pointed out, the advantage comes from the client local update. The client needs local update to find a good binary neural network. If we set the local update step size to $1$, the search of the binary weight has not started and the relatively large quantization error will cause divergence.
>
> Reisizadeh et al. FedPAQ: A Communication-Efficient Federated Learning Method with Periodic Averaging and Quantization. AISTATS 2020.
>
> 2. > 2.1 I think this paper should at least mention these works, and also compare the performance.
>
>     We thank the reviewer for the suggestion on related literature in distributed learning.
> We would like to note that our focus is on federated learning, where the server does not have access to client data.
> Therefore, DRACO[Chen’18], DETOX[Rajput’19], Election coding [Sohn’20], and ByzShield [Konstantinidis’21], which studied the redundant computation in distributed learning, where the server assigns duplicate data/tasks to workers to enhance Byzantine resilience, are not closely related to our work and can provide limited insights. For the recent arXiv paper SignGuard[Xu’21], the details on their method design are not very clear (e.g., in section IV.B, the vulnerability to the "sign-flip" attack, the setting of the mean-shift algorithm). After assessing the priority, we choose not to include these papers.
>
>     > 2.2 There have been naive methods of using median or mean of median schemes for tolerating Byzantines, starting from 2017, e.g., KRUM, multi-KRUM. I guess this paper didn’t compare with these naive methods also.
>
>     In the original manuscript, we compared with the median based method in Table 1.
> KRUM assumed that the number of attackers are known, which is not applicable in our study.
> The additional information of attackers has the power to further improve the design of Byzantine-resilient schemes, including FedVote, so the comparsion with KRUM is unfair.
>
>     > 2.3 After the [Bernstein’18], there have been upcoming works [Karimireddy’19] suggesting that error feedback fixes the issue of SignSGD. I’m not sure whether the authors compared their work with this scheme.
>
>     [Karimireddy’19] presented the scheme for a single worker.
> To the best of our knowledge, extending the error-feedback signSGD to a general federated learning setting is still an open research topic and requires nontrivial design.
> This is the reason that we do not compare with signSGD with error feedback in this work.
>
>     > 2.4 So in general, I thought that the literature search and comparison is bit weak.
> SignSGD is 3-year old paper, and there have been many other papers using voting-based aggregation for Byzantine tolerance.
>
>     We thank the reviewer for the suggestion on the literature. In this work, our main focus is to show that the design of quantizing model weight is more efficient than quantizing gradient in federated learning. The Byzantine-tolerant schemes should be studied in the context of the federated setting, i.e., the server does not have access to worker data and additional knowledge on Byzantine adversaries. The works suggested by the reviewer, as we replied earlier, are in the distributed learning setting hence not closely related to our immediate research questions, and we therefore choose not to include them.

---

### Official Review · Reviewer_LKNM · 2021-11-03

**Correctness:** 3
**Technical Novelty And Significance:** 3
**Empirical Novelty And Significance:** 3
**Recommendation:** 6
**Confidence:** 4

**Main Review:**

Strengths:

1) The paper presents a novel communication efficient federated learning algorithm that quantizes local model weights as opposed to quantizing the local gradients. The paper employs the existing approach of quantizing normalized versions of latent weights to achieve this objective.
2) The paper address the issue of Byzantine clients and proposes a weighted aggregation method to counter such Byzantines.
3) Theoretical and empirical results in the paper showcase the utility of the proposed algorithm.

Weaknesses:

1) What purpose does Lemma 1 (one-shot FedVote) serve? The assumption that all the error events are i.i.d. would most like not to hold in a multi-round setting. Even in a single round, would one need independent initializations at different clients for the analysis to hold? Also, why can the probabilities $epsilon_{m, i}$ not be arbitrarily close to 1.

2) Section 4.3 states that (without weighted aggregation) the proposed method is going to be vulnerable to the presence of Byzantines as (on average) it behaves as the FedAvg (Lemma 2). Is this conclusion correct even when one is ensuring weight normalization at the clients?

3) In Section 6, what is the underlying quantization rule when sending ternary weights to the server.

4) Please make the plot colors consistent across Fig 1(a) - 1(c).

5) In the introduction the authors claim that "However, directly quantizing the gradient vector does not provide the optimal trade-off between communication efficiency and model accuracy."  Is there any prior work that supports this blanket statement?

**Summary Of The Paper:**

This paper aims to reduce the communication cost for federated learning while ensuring that the aggregation method at the server is tolerant to the presence of Byzantine clients. Towards this instead of communicating the local update/gradients or their quantized version, this paper proposes to communicate the quantized model weights to the server. The server performs the aggregation in the quantized space and communicates the aggregate value back to the clients. To enable faithful quantization of the weights during the communication with the server, each client learns normalized weights (by applying a coordinate-wise normalization function to the latent weights).

The paper also proposes a weighted aggregation mechanism at the server that takes the reputations of different clients into account.

The paper analyzes the impact of the weight quantization on the convergence of the underlying federated learning algorithm in an i.i.d. setting. Empirical evaluations on Fashion-MNIST and CIFAR-10 show that the proposed method outperforms existing local update/gradient compression-based schemes.


**Summary Of The Review:**

The studies an important problem in the context of federated learning, namely designing communication efficient learning algorithms in the presence of Byzantine clients. The paper presents an interesting weight quantization-based algorithm and analyzes its convergence in an i.i.d. setting. The authors also demonstrate the superiority of the proposed algorithm on existing communication efficient federated learning algorithms in the literature on two standard image classification benchmarks.

That said, there are some questions that remain about the claims made in the paper (see weaknesses above).

---

> ### Author Response · Authors · 2021-11-20
> **Response to Reviewer LKNM Part 1**
>
> 1. We thank the reviewer for raising this question.
> > What purpose does Lemma 1 (one-shot FedVote) serve?
>
>     Lemma 1 shows that each bit that participated in plurality voting contributes to the estimation of the optimal weight, and conveys more information compared to other gradient quantization methods.
>
>     > The assumption that all the error events are i.i.d. would most like not to hold in a multi-round setting. Even in a single round, would one need independent initializations at different clients for the analysis to hold?
>
>     No, mainly because different workers make mistakes independently. Even though workers start from the same model received from the server at the beginning of each communication round, they work on their own local datasets and do not communicate with each other. This resembles a scenario of students learning from a teacher but making independent mistakes provided that they do not communicate with each other. We therefore have assumed independent but not necessarily identically distributed error probabilities.
>
>     > Also, why can the probabilities $\epsilon_{m,i}$ not be arbitrarily close to 1?
>
>     In both the original manuscript and the revised manuscript (although with minor notational changes),
> We do allow individual error $\epsilon_{m,i} \in [0, 1]$ but restrict the averaged error probability across all workers, $\frac{1}{M} \sum_{m=1}^m \epsilon_{m,i}$, to be upper bounded by $\frac{1}{2}$. The latter assumption for the overall error is the least stringent for the upper bound of Lemma 1 to hold, and is empirically justified by the fact that FedVote learning algorithm converges in practice.
> We have modified Lemma 1 to avoid confusion:
> - Suppose the error events $\\{w^{(k, \tau)}\_{m, i} \neq w_{i}^{*}\\}\_{m=1}^M$  are mutually independent, and the mean error probability $s_i = \frac{1}{M}\sum_{m=1}^M \varepsilon_{m,i}$ is smaller than $\frac{1}{2}$.
>
> 2. > Is this conclusion correct even when one is ensuring weight normalization at the clients?
>
>     Yes, this is a correct observation.
> The server will use the collected votes to build the empirical weight distribution and build the latent weight $\boldsymbol{h}$ [see Eqs. (13)--(14)].
> Adversaries sending the opposite results will evidently negatively affect the estimation of the weight distribution and impede the convergence in multiple rounds.
> We have added the experimental results in Fig 6 in Appendix C.2 to show the vulnerability of the original FedVote when there exist Byzantine attackers.
>
> 3. It is still based on the stochastic quantization similar to the binary case in (11).
> We have added the following details in Appendix C.3 of the revised manuscript to avoid ambiguity: \\[10pt]
> The stochastic rounding used in the ternary neural networks, $w_i = Q_{\text{sr}}(\widetilde{w})$, is an extension of (11):
> \begin{equation}
>     w_i = \left\\{
> \begin{array}{ll}
>    +1, & \textrm { with probability } \pi\_{1} = \tilde{w}_i  \mathbf{1}(\widetilde{w}_i > 0),   \\\\
>    -1, & \textrm{ with probability } \pi\_{2} = - \tilde{w}_i  \mathbf{1}(\widetilde{w}_i < 0),  \\\\
>    0, & \textrm{ with probability } 1 - (\pi\_{1} + \pi\_{2}).
> \end{array}\right.
> \end{equation}
>
> 4. > Please make the plot colors consistent across Fig 1(a) - 1(c).
>
>     We thank the reviewers for the suggestion.
> We have updated the manuscript using unique and consistent color--marker patterns for Fig 4(a)~(c).
>
> 5. > Is there any prior work that supports this blanket statement?
>
>     We thank the reviewer for raising this question.
> Since this hasn't been verified rigorously in prior works, we modified the sentence to:
>   - However, directly quantizing the gradient vector \textbf{may not} provide the optimal trade-off$^1$...''
>
> $^1$Predictive coding in video and image compression (Li et al., 2015; Gonzalez et al., 2014) is an example that directly quantizing the raw signal we intend to transmit does not provide the best trade-off between the coding efficiency and the utility/bitrate.
>
> Li et al. ``Lagrangian Multiplier Adaptation for Rate-distortion Optimization with Inter-frame Dependency." IEEE Transactions on Circuits and Systems for Video Technology 2015.
>
> Gonzalez et al. ``Digital Image Processing.''  Third Edition  2014.

---

> > ### Comment · Reviewer_LKNM · 2021-11-29
> > **Keeping my original score**
> >
> > Thank you for your response. After going through the response and other reviews, I have decided to keep my original score.

---

### Official Review · Reviewer_mhMk · 2021-11-07

**Correctness:** 3
**Technical Novelty And Significance:** 2
**Empirical Novelty And Significance:** 2
**Recommendation:** 3
**Confidence:** 4

**Main Review:**

Mainly, the paper has the following weaknesses:

- The main proposed method (FedVote) is incremental at best, and the proposal on the adversarial aspect (Byzantine-FedVote) is limited. The normalization mechanism potentially requires careful tuning. Binary weights reduce uplink communication, but downlink savings depend on probability quantization scheme.

- Confined to using binary neural networks

- It is not clear how the quantization of probability values for soft voting are set (i.e., predefined thresholds) and the effects on performance.

- It is not clear how the predefined coefficients ($\beta$) are set as it relates to reputation-based voting, and the effects on performance.

- Empirical evaluations is limited:
  - Several other machine learning models/benchmarks are not evaluated on (e.g., [LEAF: A Benchmark for Federated Settings, Caldas et al. 2019]).
  - For test accuracy experiments, the number of clients used and the number sampled each round is not given.
  - For communication efficiency experiment, the number of clients used ($M$=31) is quite small (e.g., [Hsu et al. 2019] uses 100 clients). Down-link GB not accounted for.

- Preliminaries should be provided on adversarial FL/Byzantine attack.

- Information on measuring energy usage (Table 4) not provided.

Are Figure 4 (a) and (b) correctly labeled?


**Summary Of The Paper:**

The paper presents a scheme for federated learning (FL) called FedVote. In FedVote, clients use binary neural networks where a latent restricted range weight vector $\boldsymbol{h}$ is learned, and stochastic rounding is applied to obtain quantized weights $\in$ {$-1,+1$}. The communication cost to the server (up-link) is reduced by using quantized weights. The server aggregates weights by summing over all the weights of the clients at each round of training and applying a sign function. This mechanism is referred to as plurality voting (ties are broken randomly). However, the paper also proposes to use soft voting, which normalizes the number of ones (cf. hard sign function) to get probability values. These probabilities are then quantized with a clipping mechanism to restricted maximum and minimum values, using predefined thresholds, and broadcasted to the clients (down-link). The latent weights of the client models are updated using the soft voting results. The paper also adapts work on reputation-based voting (Bendahmane et al. 2014) to deal with adversarial FL in the form of Byzantine attacks. The paper presents convergence analysis for FedVote under the independent and identically distributed data setting. Empirical evaluations is conducted on two datasets ,Fashion-MNIST and CIFAR-10, and applied to two models, LeNet-5 and VGG-7, respectively.


**Summary Of The Review:**

The contributions in the paper are limited and not well-focused. Overall, I do not think the paper is at a level for acceptance.

---

> ### Author Response · Authors · 2021-11-20
> **Response to Reviewer mhMk Part 1**
>
> 1. We thank the reviewer for raising these concerns. Below we address them point by point.
> > The main proposed method is incremental at best.
>
>     We respectfully disagree with the reviewer's assessment. To the best of our knowledge, this is the first paper systematically studying the quantized neural network optimization in FL. We showed that directly quantizing the gradient may not provide the best tradeoff between communication efficiency and model accuracy. In comparison, quantizing neural network weight in our design is a better choice. This result has not been discussed in the literature before. Since most of the frameworks focused on reducing the size of gradient, we hope this work can diversify the design choices of communication-efficient FL algorithms.
>
>     > The normalization mechanism potentially requires careful tuning.
>
>     We thank the reviewer for raising the concern on the reproducibility. The normalization function $\varphi(x) = \tanh(ax)$ has only one scalar parameter, $a$. In Section 6 of the original manuscript, we had a discussion on the choice of $a$ and gave its default value $3/2$.
>
>     > Binary weights reduce uplink communication, but downlink savings depend on probability quantization scheme.
>
>     The uplink communication cost is the main focus in this work. We have added the following sentence in the first paragraph of Section 4:
> - We follow the widely adopted analysis framework in wireless communication to investigate only the worker uplink overhead, assuming that the downlink bandwidth is much larger and the server will have enough transmission power [1].
>
>     In addition, some recent FL algorithms also put their main focus on the uplink communication cost, such as FedPAQ [2] and FedCOM [3].
>
> [1] Tran et al. Federated Learning Over Wireless Networks: Optimization Model Design and Analysis. INFOCOM 2019.
>
> [2] Reisizadeh et al. FedPAQ: A Communication-Efficient Federated Learning Method with Periodic Averaging and Quantization. AISTATS 2020.
>
> [3] Haddadpour et al. Federated Learning with Compression: Unified Analysis and Sharp Guarantees. AISTATS 2021.
>
> 2. > Confined to using binary neural networks
>
> In Section 6 of the original manuscript,  we presented the extension to the ternary neural networks and reported the test accuracy.
> We noted in the paper that the method designed for the binary neural network can be extended to neural networks with more quantization levels by using different normalization functions.
>
> 3. > It is not clear how the quantization of probability values for soft voting are set and the effects on performance.
>
> We thank the reviewer's question to improve the reproducibility and readability of our paper.
> We have added the following sentences in Appendix C.1:
> - For the clipping thresholds, we set $p_{\text{min}} = 0.001$ and $p_{\text{max}} = 1 - p_{\text{min}}$.
> The thresholds are introduced for numerical stability and have little impact on performance.
>
> 4. > It is not clear how the predefined coefficients ($\beta$) are set as it relates to reputation-based voting, and the effects on performance.
>
> Thanks. We have added the following sentence Appendix C.1:
> - We use $\beta = 0.5$ in Byzantine-FedVote. The choice of the smoothing factor $\beta$ has little impact on the final test accuracy, as the credibility score decays exponentially for adversaries over multiple communication rounds.
>
> 5. > Several other machine learning models/benchmarks are not evaluated on
>
>     The Dirichlet non-i.i.d. partition of CIFAR-10 dataset is a widely-adopted benchmark in the literature [4-7]. We want to point out that our goal is to show the effectiveness of quantizing model weight in communication-efficient federated learning and we do not target at outperforming state-of-the-art algorithms on non-i.i.d. federated datasets.
> We believe that additional benchmarking is not necessary.
>
>     > For test accuracy experiments, the number of clients used and the number sampled each round is not given.
>
>     In the paragraph "Communication Efficiency and Convergence Rate" of Section 6 of the original manuscript, we wrote "we consider $M=31$ workers." We have modified this sentence to avoid confusion:
> - we consider $M=31$ workers with full participation.
>
>     > For communication efficiency experiment, the number of clients used (=31) is quite small.
>
>     We have increased the number of clients to 100 with partial participation and added the results in Appendix C.3.
>
>     > Down-link GB not accounted for
>
>     Please see our response to the comments on "downlink savings".
>
> [4] Hsu et al. Measuring the Effects of Non-identical Data Distribution for Federated Visual Classification. arXiv 2019.
>
> [5] Chen et al. FedBE: Making Bayesian Model Ensemble Applicable to Federated Learning. ICLR 2021.
>
> [6] Yoon et al. FedMix: Approximation of Mixup Under Mean Augmented Federated Learning. ICLR 2021.
>
> [7] Acar et al. Federated Learning Based on Dynamic Regularization. ICLR 2021.

---

> > ### Author Response · Authors · 2021-11-20
> > **Response to Reviewer mhMk Part 2**
> >
> > 6. > Preliminaries should be provided on adversarial FL/Byzantine attack.
> >
> >     We thank the reviewer for the suggestion.
> > In our original manuscript, we had the literature review part for Byzantine attacks. Additionally, in the "Byzantine Resilience" paragraph of Section 6, we further elaborated that "We consider omniscient attackers who can access the datasets of normal workers and send the opposite of the aggregated results to the server." We believe that the context is clear and more preliminaries on Byzantine attacks are not necessary.
> >
> > 7. > Information on measuring energy usage (Table 4) not provided.
> >
> >     Thanks for raising the clarification concern. In the revised version, we have improved the presentation by explicitly providing per the operation costs of multiplication and addition, respectively. We added the following sentence to the "Deployment Efficiency" paragraph:
> >     - As to the energy consumption calculation, we use $3.7$ pJ and $0.9$ pJ as in Hubara et al. (2016) for each floating-point multiplication and addition, respectively.
> >
> > Hubara et al. Binarized Neural Networks. NeurIPS 2016.
> >
> > 8. > Are Figure 4 (a) and (b) correctly labeled?
> >
> >     Yes, they are correctly labeled but the color schemes used were different from Figure 4(c).
> > We have updated the manuscript using unique and consistent color--marker patterns for Fig 4(a)--(c).

---

> > > ### Comment · Reviewer_mhMk · 2021-11-30
> > > **Maintaining recommendation**
> > >
> > > I want to first thank the author(s) for their responses and effort on revising the paper. Also, I have read through the other reviews and responses.  However, the responses (and updates) do not fully address the issues I brought up, and thus, the paper is still not at the level of acceptance for ICLR. I maintain my original score.

---

### Decision · Program_Chairs · 2022-01-20

**Decision:**

Reject

**Comment:**

Reviewers raised several valid concerns about novelty of quantization idea and lack of discussions related to prior art (AISTATS 2020 paper). The rebuttal did not convince the reviewers to raise their score. We hope the authors will benefit from the feedback and improve the paper for future submission.